# *Neisseria meningitidis* filamentous phage MDA promotes colonisation by selecting hyperadhesive pili variants

Clémence Mouville[1], Antoine Brizard[1], Morgane Wuckelt [1], Mélanie Montabord[1], Hervé Lécuyer[1,2], Julie Meyer [1], Anne Jamet [1,2], Béatrice Durel[3], Charlotte Izabelle[4], Xavier Nassif[1,2], Mathieu Coureuil [1] ✉ & Emmanuelle Bille [1,2] ✉

Filamentous phages are non-lytic phages mutually beneficial to their bacterial hosts. In *Neisseria meningitidis*, the filamentous phage MDA is associated with invasive diseases thanks to its key role in the formation of biofilm during epithelium colonisation. The infection model for filamentous phages has been defined for phages Ff and CTX. These phages bind to the tips of bacterial pili before being translocated into the periplasm of their hosts. The aim of this study is to investigate the relationship between filamentous phage infection and type IV pili, using the bacterium *Neisseria meningitidis* and the bacteriophage MDA as model organisms. We show that MDAΦ rather binds to type IV pili along their entire length with preferential binding to positively charged variants of the major fibre-forming pilin, demonstrating a role for antigenic variation in phage infection. Strikingly, bacteria expressing the more positively charged pilin are also the most adhesive, suggesting that MDAΦ primarily target the most adhesive bacteria. Finally, we show that adhesion to human cells is sufficient to amplify the phage-positive meningococcal population. Overall, this study reveals how a filamentous phage can target hyperadhesive bacterial variants and promote their selection, thereby establishing a link between phage infection and bacterial colonisation.

The contribution of phages to the virulence of their bacterial hosts has been studied mainly for *Caudovirales*, which represent 96% of the known phages[1]. In contrast, little is known about inoviruses, a family of bacteriophages carrying a single-stranded circular DNA with unique morphology and infectious cycle characteristics, to which the filamentous phage subfamily belongs. However, filamentous phages are found in several major pathogenic species[2]. The first filamentous phage of this family was described in *Escherichia coli* in the 1960s by several groups[3–6]. A second well-known filamentous phage, hosted by *Vibrio cholerae*, was discovered by Matthew K. Waldor and John J. Mekalanos. This phage, called CTXΦ, encodes the cholera toxin genes, which promotes the dissemination of CTX-carrying *V. cholerae*[7]. The main feature is the absence of bacteria lysis during the infectious cycle[8]. The Pf4 phage of *Pseudomonas aeruginosa* is another example. It remains inactive until the bacteria perceive environmental signals (such as oxidative stress), then Pf4 genes are among the most induced genes in *P. aeruginosa* biofilms where Pf4 filamentous phages contribute to biofilms as a structural components and as a physical barrier

[1]Institut Necker-Enfants Malades, INSERM U1151, CNRS UMR8253, Université Paris Cité, Paris, France. [2]Department of Clinical Microbiology, Assistance Publique-Hôpitaux de Paris (AP-HP), Hôpital Necker-Enfants Malades, Paris, France. [3]Cell Imaging Platform, Structure Fédérative de Recherche Necker INSERM US24/CNRS, UMS3633, Paris, France. [4]Faculté de Pharmacie de Paris, Plateforme d'Imagerie Cellulaire et MOléculaire PICMO, US25 Inserm, UAR3612 CNRS, Université Paris Cité, Paris, France. ✉e-mail: mathieu.coureuil@inserm.fr; emmanuelle.bille@inserm.fr

to antibiotics and as an important immune modulator during infection[9]. Finally, filamentous phages have been described in *Neisseria* species and in particular in *Neisseria meningitidis* where the filamentous phage named MDA for Meningococcal Disease Associated phage, also known as Nf1 phage[10], plays a key role in biofilm formation and is associated with virulence[11–14]. Other filamentous bacteriophages contribute to the virulence of many bacteria in a variety of ways[2].

Filamentous phages that infect diderm bacteria must cross both the outer and inner membranes to infect their target. Although the cell surface receptors are unknown for the majority of filamentous phages described so far, a role for retractable bacterial pili has been suspected for their entry, such as Ff and related phages that use the sexual/conjugative pilus F of *E. coli*[2,15]. Early observations of f1 phages by electron microscopy suggested that the phages bind to the tip of the pilus[15]. This was confirmed experimentally for CTXΦ, in which the phage adsorption protein pIII interacted directly with the pilus tip-subunit TcpB[16]. In the case of *N. meningitidis*, we have previously shown that MDA phage entry into the host requires retractable type IV pili[17], but the bacterial proteins involved in the MDAΦ-pili interaction are not known.

Thanks to pili retraction, the phage is thought to cross the bacterium's outer membrane through the pilus secretin and then enter the periplasm, where, as previously described, the Ff or CTX phages interact with the TolQRA system for the disassembly and entry of the single-stranded DNA viral genome into the cytosol. In the absence of the TolQR complex, the homologous ExbBD complex can perform phage uptake with TolA[18,19]. Overall, our knowledge of the molecular mechanisms by which filamentous phages interact with their associated pilus and enter bacteria is incomplete for most filamentous phages and remains to be determined.

*N. meningitidis* is a commensal bacterium, commonly present asymptomatically in the human nasopharynx. Under certain circumstances, the bacterium can enter the bloodstream and cause sepsis and/or meningitis after crossing the blood-brain barrier[20]. Survival in the extracellular fluid is dependent on a small number of virulence factors such as the capsular polysaccharide, the iron chelation systems and factor H binding protein[21,22]. In addition, type IV pili are essential for blood vessel colonisation, a key step in the pathogenesis of *N. meningitidis*[23–26]. Other bacterial attributes, such as Opa proteins, are likely to play an essential role in nasopharyngeal colonisation[27]. While few virulence factors are associated with meningococcal invasiveness, no virulence factor provides a clear explanation for why certain strains of *N. meningitidis* are responsible for outbreaks of invasive infection. One exception is the presence of an 8 kb island in the genome of invasive isolates, the meningococcal filamentous MDA prophage[10,12–14,17]. The MDA prophage may be found up to 4 times per genome, with partial heterogeneity in these sequences suggesting that a strain can be re-infected by the same phage or by a phage from another strain[28,29]. Previous results have suggested that MDA phages enhance epithelial cell colonisation by promoting bacterial-bacterial interactions and increasing bacterial biomass in the nasopharynx[11], which in turn should increase the frequency of bacterial dissemination in the bloodstream and/or the spread of bacteria within the human population.

The dissemination of the MDA phage is likely to be associated with the hijacking of the type IV pilus machinery, a feature that appears to be shared with most other inoviruses such as CTX and Pf4. Type IV pili are highly dynamic filaments that generate tensive forces during cycles of elongation and retraction, powered by a dedicated elongation motor called PilF and retraction motor called PilT[30]. Filament biogenesis relies on a complex multi-protein machinery that, in *N. meningitidis*, contains 15 proteins essential for biogenesis (PilC1 and C2, D, E, F, G, H, I, J, K, M, N, O, P, Q, W), PilE being the pili-forming pilin and PilQ the secretin through which the pilus crosses the outer membrane. As determined by cryo-electron tomography, the type IV pili biogenesis

machinery is a multilayered structure spanning the entire cell envelope[31,32]. Meningococcal strains express either a class I or a class II *pilE* gene[33]. Class I *pilE* are subject to antigenic variation and type IV pili are therefore considered to be one of the most variable structures on the meningococcal surface. This *pilE* gene consists of a constant region and a variable domain that can vary by recombination with one of the eight *pilS* silent cassettes. In addition, in the early 1990s, it was shown that antigenic variation in PilE plays an important role in cell adhesion to the host and escape from the immune response[34,35]. Conversely, the class II pilins do not undergo antigenic variation and form a class of conserved pilin subunits[36]. In addition, the PilE protein is highly post-translationally modified (glycosylation and addition of phosphoglycerol)[37]. Finally, type IV pili have been implicated in numerous functions including bacterial interaction with human cells, DNA uptake, twitching motility and bacterial-bacterial interactions[34,38–40].

Here, we use *N. meningitidis* as a model to dissect the molecular mechanism of the interaction between filamentous phages and type IV pili, and how this shapes the bacterial population. Unlike most filamentous phages, we demonstrate that the type IV pili tip adhesin is not involved in infection. Conversely, we observe the binding of MDAΦ to type IV pili along their fibre, an interaction that took advantages of the positive charges carried by certain PilE variants that are also strongly associated with type IV pili-mediated adhesion, suggesting a key role for antigenic variation in MDAΦ infection. Finally, we show that adhesion to human cells is sufficient to allow enrichment of a meningococcal population with phage-positive bacteria, demonstrating the association between adhesion and the presence of the MDAΦ, two features associated with the invasiveness of *N. meningitidis*.

## Results
### MDAΦ infection is dependent on retractable type IV pili and specific *pilE* sequences
Interaction of inoviruses with their host bacteria and subsequent entry into the cytosol is thought to be highly pili-dependent. In the case of MDAΦ, we have previously shown that both the major pilin PilE and the retraction ATPase PilT are required for phage transduction, whereas other minor pilins of the fibre (PilV, ComP, and PilX) are not required[17]. To gain further insight into the pili-phage interaction, we extended our previous screen by assessing the efficacy of phage transduction in a comprehensive library of individual mutants of the type IV pili machinery, including proteins involved in fibre assembly or in the machinery itself, (PilD, F, G, P, Q, T, T2, U, W, Z and TfpC, TsaP and NMA0415) and proteins that compose the fibre (PilE, PilV, PilX, ComP) or the tip (PilH, I, J, K, C1, and C2) (Fig. 1a–d). Transduction efficiency was quantified in a strain deleted for the MDA prophage (ZΔ*MDA*) and using a purified MDA phage carrying a spectinomycin resistance gene (MDA$_{(orf6-aadA1)}$Φ) for colony forming unit (CFU) quantification (Fig. 1a). Note that the ZΔ*MDA* strain is derived from a clinical isolate that has not been selected for a specific *pilE* sequence - as for the NEM8013 2C4.3 strain (clone 12)[41] so it corresponds to a mix of *pilE* sequences. Transduction was compared with transformation efficiency (Fig. 1b) and pili quantity (Fig. 1c). The pili quantity was estimated by immunostaining of the *pil* mutants introduced into a ZΔMDA *pilE$_{SB}$* expressing strain[34]. This strain expresses type IV pili, which can be recognised by the 20D9 antibody[42]. Transformation and pili quantity are two proxies for the expression of functional type IV pili[43]. Transduction was impaired in mutants with low pili quantity and low transformation efficiency, as well as in the Δ*pilT* mutant, confirming that the phage only infects strains expressing fully retractable pili. The contribution of the proteins thought to form the pilus tip (PilH, I, J, K) is difficult to interpret as their respective mutants were very poorly piliated. However, the infection efficiency of the four individual mutants was higher than that of the non-piliated Δ*pilE* mutant, suggesting that these proteins have a minor role in phage infection.

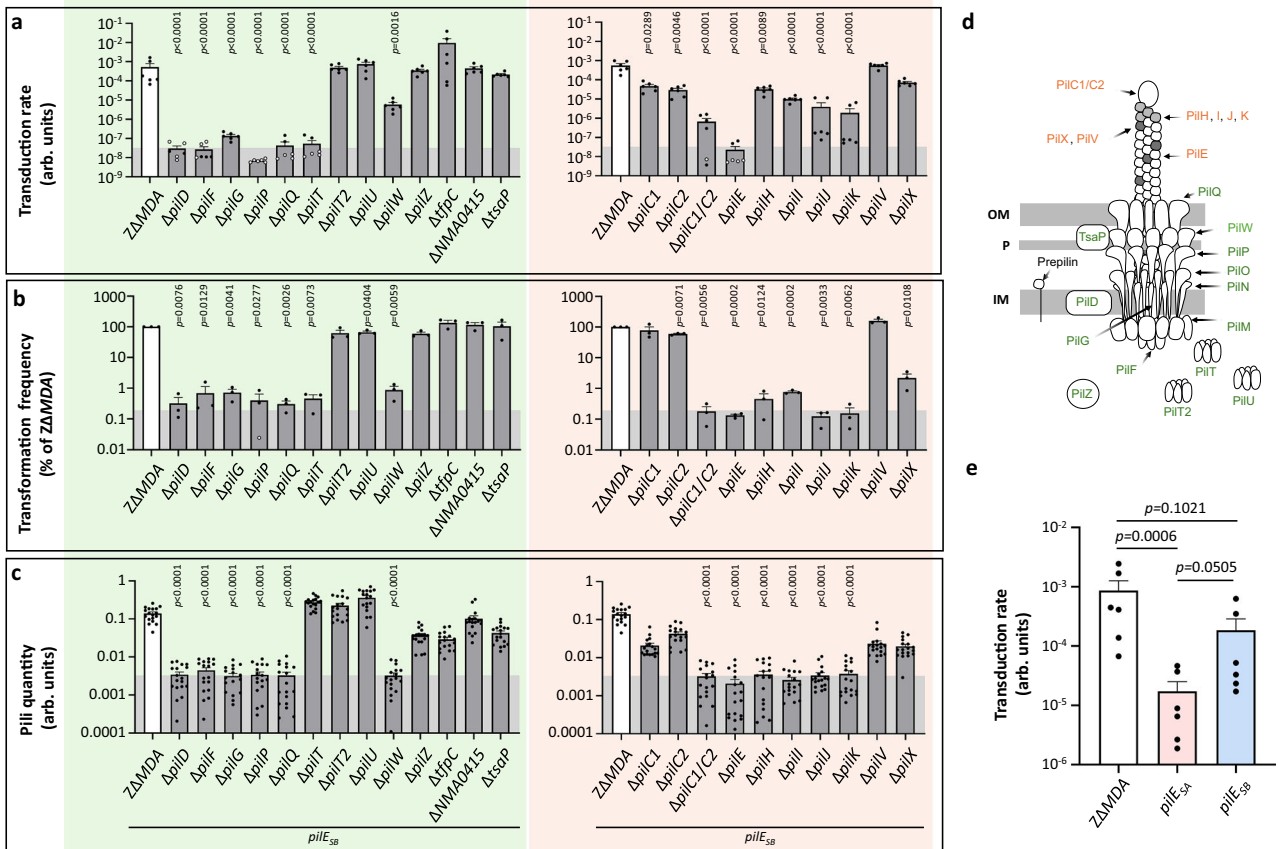

**Fig. 1 | Involvement of type IV pilus machinery and PilE variants in MDA phage infection.** Phenotypes of transduction (**a**), transformation (**b**) and pili quantity (**c**) for the ZΔ*MDA* reference strain and its derivative mutants in the pilus machinery genes (green) and pilus fibre genes (orange). Transduction results (**a**) are expressed as transduction rates (arb. units: arbitrary unit), transformation frequency results (**b**) as percentages of the reference strain ZΔ*MDA* and pili quantity as raw data (arb. units: arbitrary unit). For transformation and transduction experiments, it should be noted that the background should be due to spontaneous appearance of resistant clones. White-centered dots correspond to CFU below the detection limit of respective experiments (see Methods section). The detection threshold is visualized as a shaded grey area, it corresponds to the frequency of spontaneous resistance to spectinomycin. Exact *p*-values are provided in the Source Data file. Transduction assays were performed three times (*n* = 3) in duplicate and statistical analyses were performed using a Brown–Forsythe and Welch ANOVA test (two-

sided) with Holm–Sidak's correction for mutants in the pilus machinery genes and using Ordinary one-way ANOVA test with Dunnett's correction for mutants in the pilus fiber genes. Transformation frequency assays were performed independently three times (*n* = 3), and statistical analyses were performed using a One-sample *t*-test (two-sided) comparing the sample mean with a hypothetical mean of 2. All data were expressed as mean ± SEM. Pili quantifications were performed in triplicate (*n* = 3) and statistical analyses were performed using a Kruskal–Wallis test with Dunn's correction. Source data are provided as a Source Data file. **d** Mutated proteins are indicated on the schematic representation of the type IV pilus machinery. **e** Transduction rate for the ZΔ*MDA* reference strain and derivatives expressing the PilE$_{SA}$ or PilE$_{SB}$ variants. Transduction assays were performed three times (*n* = 3) in duplicate and statistical analyses were performed using an Ordinary one-way ANOVA test with Tukey correction, and data were expressed as mean ± SEM. Source data are provided as a Source Data file.

Finally, we studied PilC1 and PilC2, two proteins that are thought to be located at the tip of the type IV pilus based on homology with other type IV pilus models[44–46]. The Δ*pilC1/C2* double mutant is barely piliated and does not support efficient phage transduction. Conversely, both Δ*pilC1* and Δ*pilC2* single mutants were piliated and fully transduced by the phage (Fig. 1a).

Finally, we investigated the role of PilE in phage entry and transduction. We took advantage of the antigenic variation of the class I *pilE* gene and analysed the transduction efficiency of MDA$_{(orf6-aadA1)}$Φ depending on *pilE* sequence. We took advantage of previous constructs available in our team in which *pilE$_{SA}$* and *pilE$_{SB}$* variant sequences were fused to a kanamycin resistance gene to allow their selection[34]. We therefore constructed two phage-receptive strains with the two different *pilE* variant sequences[34]. The two strains differ in the variable part of PilE and especially in the D-region between the two conserved C-terminal cysteine residues (C$_{120}$ and C$_{152}$ for PilE$_{SA}$) (Supplementary Fig. 1a–c). We then evaluated the transduction efficiency in these two strains compared to the parental strain ZΔ*MDA*. The MDA$_{(orf6-aadA1)}$Φ transduction efficiencies differ among the

*pilE$_{SA}$*, *pilE$_{SB}$*, and ZΔ*MDA* strains. The *pilE$_{SA}$* strain exhibits a significantly reduced phage transduction rate compared to ZΔ*MDA* (Fig. 1e, *p* = 0.0006). The transduction rate of *pilE$_{SA}$* is also lower than that of *pilE$_{SB}$*, although this difference is not statistically significant (*p* = 0.0505). These findings suggest a potential role of the *pilE* sequence in modulating transduction efficiency. Finally, we investigated the role of PilE post-translational modification in phage transduction (Supplementary Fig. 1d). Mutations in the two genes encoding the glycosyltransferases PglA and PglC or in the gene encoding pilin phosphotransferase B (PptB) have little effect on phage transduction.

## PilS loci are a reservoir of pili sequences compatible with MDAΦ infection

Our results above suggested that PilE variants may have different efficiencies for infection by MDAΦ. However, the Fig. 1e raised the question of whether transduced meningococci from the PilE$_{SA}$-expressing strain still express this PilE variant or whether the phage has infected bacteria expressing another PilE variant that is present in the

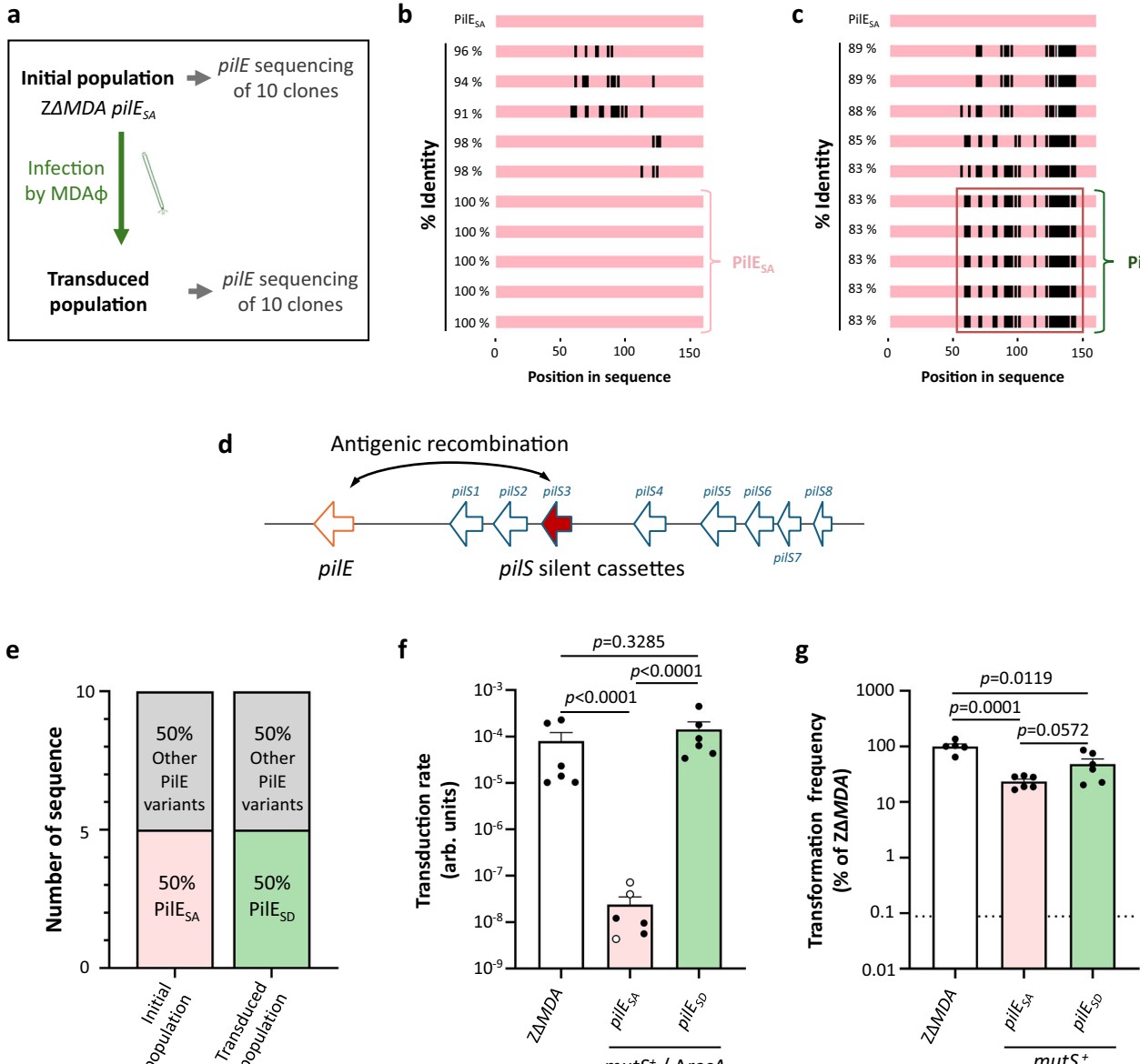

**Fig. 2 | Antigenic variation of PilE impacts phage transduction. a** Schematic representation of the methodology used in (**b**, **c**, **e**). The *pilE* gene of ten clones randomly selected from the population before and after transduction by MDA_(orf6-aadA1)Φ and selection on antibiotic was sequenced. **b–e** The panels depict the amino acid sequence alignments deduced from the sequenced *pilE* genes of the 10 clones obtained before selection (**b**) and after (**c**) selection. The PilE_SA variant was used as a reference and the regions identical to PilE_SA were coloured in pink. Gaps were coloured in white and mismatched regions were coloured in black. The percentage of identity relative to the PilE_SA sequence is indicated to the left of each sequence. **d** A schematic representation of the *pilE* gene locus and its eight silent *pilS* cassettes involved in the antigenic recombination of PilE. The *pilS3* cassette, in red, corresponds to the sequence found in the variants framed in red in

(**c**). **e** A stacked bar plot showing the distribution of the PilE variant relative to the whole population before and after selection. Phenotypes of transduction (**f**) and transformation (**g**) for the strain ZΔ*MDA* and its derivatives expressing the PilE_SA or PilE_SD variants. Transduction results (**f**) were expressed as transduction rates (arb. units: arbitrary unit) and transformation results (**g**) were expressed as percentages of that of ZΔ*MDA*. Experiments were performed three times (*n* = 3) in duplicate. Statistical analyses were performed using an ordinary ANOVA test with Tukey correction, and data were expressed as mean ± SEM. White centered dots correspond to CFU below the detection limit. The dotted line corresponds to the mean threshold for the appearance of spontaneous nalidixic acid resistant clones. Source data are provided as a Source Data file.

population due to antigenic variation despite the use of the kanamycin resistance gene in frame with *pilE_SA*. To answer this question, we first sequenced the *pilE* gene of 10 independent clones of the ZΔ*MDA pilE_SA* strains (Fig. 2a, b). Seven clones expressed the PilE_SA sequence or a closely related sequence (98% identity) and three clones had more than five different mutations compared with the PilE_SA sequence (Fig. 2b). We then transduced the ZΔ*MDA pilE_SA* strain with MDA_(orf6-aadA1)Φ, selected the transduced population on spectinomycin and sequenced 10 independent clones again (Fig. 2a, c). After transduction, none of the 10 sequences obtained before transduction were recovered (Fig. 2b, c,

e). Five of the 10 sequences were identical and were designated PilE_SD. This sequence carried a D-region completely different from that of PilE_SA, as well as several other variations between G58 and D113 of PilE_SA (Fig. 2c, d and Supplementary Fig. 2a, b). Interestingly, the *pilE_SD* sequence is closely related to the *pilS3* silent locus of the parental strain (Fig. 2d and Supplementary Fig. 2a, b), suggesting that antigenic variation of PilE still occurs despite the addition of the kanamycin resistance gene in frame with *pilE*. To confirm that antigenic variation in the ZΔ*MDA pilE_SA* strain helps explain the level of infection by MDA_(orf6-aadA1)Φ, we constructed two *mutS⁺* Δ*recA* mutant ZΔ*MDA*

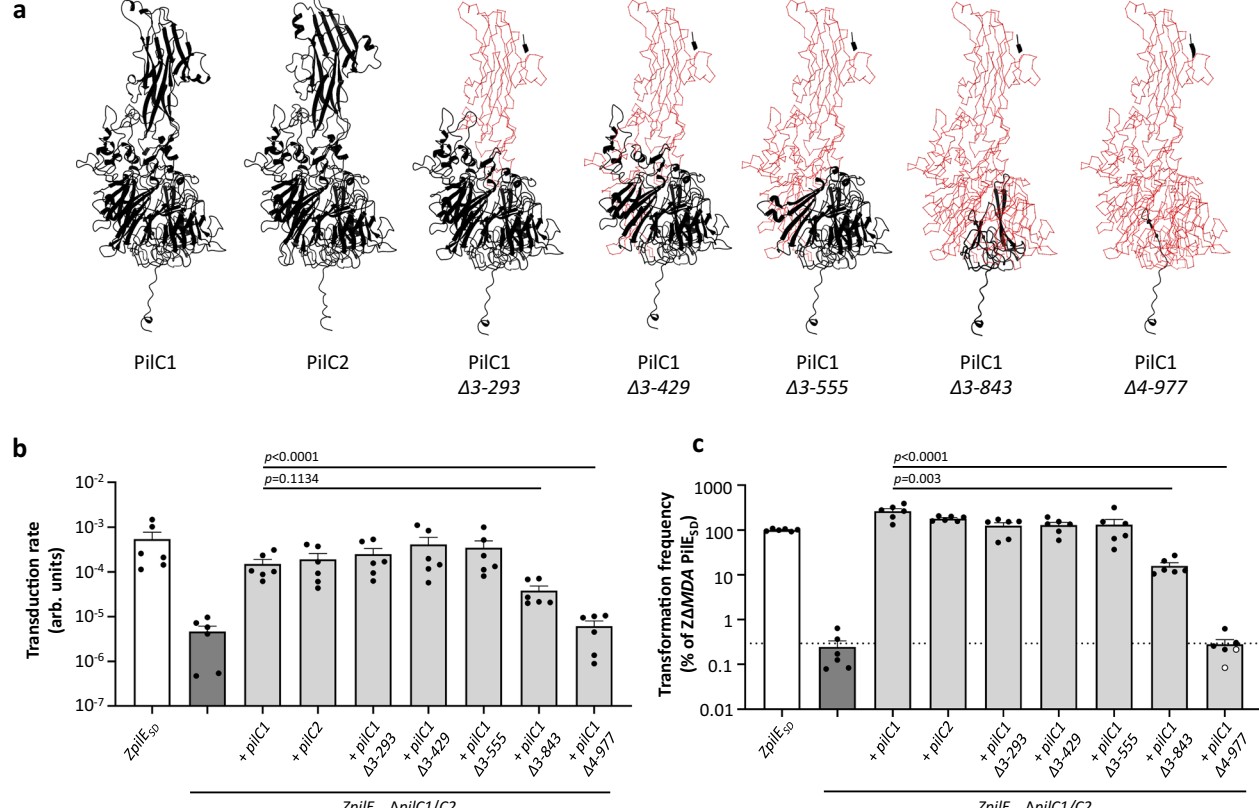

**Fig. 3 | Interaction between MDA phage and type IV pili. a** Predicted structures of PilC1 and PilC2 were obtained using Alphafold-3. The deleted parts of PilC1 were shown as a thin line in red. Phenotypes of transduction (**b**) and transformation (**c**) for the strain Z5463 expressing the PilE$_{SD}$ variant (*pilE$_{SD}$*) and its derivative double mutant Δ*pilC1/C2* complemented or not with the whole *pilC1* or *pilC2* genes or *pilC1* deletion mutants of decreasing size. Transduction results (**b**) were expressed as transduction rate (arb. units: arbitrary unit) and transformation results (**c**) were expressed as percentages of that of the strain expressing the PilE$_{SD}$ variant. Each experiment was performed three times (*n* = 3) in duplicate. Statistical analyses were performed using an ordinary ANOVA test with Dunnett correction for transduction rates, a Kruskal–Wallis test with Dunn's correction for transformation frequencies, and data were expressed as mean ± SEM. White-centered dots correspond to CFU below the detection limit. The dotted line on bar plot **c** corresponds to the mean threshold for the appearance of spontaneous nalidixic acid resistant clones. Source data are provided as a Source Data file.

strains expressing either PilE$_{SA}$ or PilE$_{SD}$ variants, and we infected both strains with MDA$_{(orf6-aadA1)}$Φ. Deletion of *recA* is sufficient to abolish the homologous recombination required for antigenic variation[47]. Meanwhile, the inactivated copy of the *mutS* gene in the ZΔ*MDA* strain was reverted to WT (*mutS$^+$*) to limit the spontaneous appearance of mutants[29]. The strain was subsequently designated ZΔ*MDA mutS$^+$ΔrecA*. Transduction rate was determined as above and compared to the initial ZΔ*MDA* strain (Fig. 2f). It should be noted that phage infection is independent of RecA. We recovered very few colonies with the ZΔ*MDA mutS$^+$ΔrecA pilE$_{SA}$* strain, whereas the ZΔ*MDA mutS$^+$ΔrecA pilE$_{SD}$* strain was transduced as efficiently as the initial ZΔ*MDA* strain. We then confirmed that the efficiency of transformation was in the same range between the *pilE$_{SA}$* and the *pilE$_{SD}$* in ZΔ*MDA mutS$^+$* strains - the *recA* deletion preventing homologous recombination - (Fig. 2g). While the transformation frequency was decreased in both strains compared to the parental strain, we observed no statistical differences between strains expressing *pilE$_{SA}$* or *pilE$_{SD}$*. Overall, these results suggest that: (i) some variants of PilE appear to be more susceptible to infection; (ii) MDAΦ is able to specifically infect a low-abundance clone expressing a specific PilE variant.

## The type IV pili tip is not essential for MDAΦ entry and transduction

The near absence of phage infection in the ZΔ*MDA mutS$^+$ΔrecA pilE$_{SA}$* strain suggests that the pili tip is not the receptor for MDAΦ. To rule out this hypothesis, we decided to further investigate the role of the

PilC proteins. The two PilC proteins are bimodular proteins with a similar C-terminal domain that is thought to be anchored at the pilus tip based on the *Myxococcus xanthus* model[45]. The N-terminal (N-ter) domain is involved in bacterial adhesion and differs between PilC1 and PilC2[48,49] (Fig. 3a). Since the Δ*pilC1/C2* double mutant strain is apparently barely piliated, we sought to complement a *pilC* null strain for the expression of recombinant PilC1 mutants of decreasing size (Fig. 3a) and to evaluate transduction and transformation in a strain expressing the PilE$_{SD}$ variant (Fig. 3b, c). Deletion of the amino acids from 3 to 555 in the N-terminal part of PilC1 did not affect transduction or transformation frequency, supporting the idea that the N-terminal domain of PilC is not recognised by the MDAΦ and is not required for pili biogenesis. Increasing deletion to 3–843 residues reduced both the transduction and the transformation frequencies of the resulting strain. Interestingly, the truncated protein PilC1$_{Δ3-843}$ allowed transduction and transformation at a level close to 10% of the control (although not statistically significant for transduction) while truncated PilC1$_{Δ4-977}$ has lost both and transduction and transformation competency (Fig. 3b, c). This suggests that PilC1$_{Δ3-843}$ and PilC1$_{Δ4-977}$ do not complement for piliation and cannot be included in our analysis. This supports the view that PilC does not play a role in MDAΦ entry. In a complementary approach, we investigated a possible interaction between the MDAΦ ORF6 (Supplementary Fig. 3a) and proteins of the pilus tip using the bacterial two-hybrid system. Based on structure and sequence homology with proteins pIII of M13 and CTXΦ pIII of CTXΦ, ORF6 was previously annotated as a phage adsorption protein, and we

showed in a prior work that ORF6 is required for MDAΦ transduction[17]. In the existing model, the adsorption protein should interact with the pili tip to cross the outer membrane. We therefore assessed the interaction of ORF6 with the N-terminal domain of PilC1 or PilC2, with the C-terminal domain of PilC1 and with the globular heads of the pilins PilH, I, J, and K (Supplementary Fig. 3b). No positive result was observed, suggesting that there is no interaction between ORF6 and the N-terminal domain of PilC1, assuming that the proteins are folded properly. Considering that expression of the first 843 amino acids of PilC is not required for phage transduction (see Fig. 3b), our results suggest that entry of the MDAΦ into meningococci relies mostly on a mechanism independent of an interaction between ORF6 and the pilus tip proteins PilC1/2, PilH, PilI, PilJ, and PilK.

## The MDAΦ binds to the pilus filament

Considering the role of antigenic variation in MDAΦ transduction, we reasoned that the MDA phage should interact directly with the pilus fibre rather than the tip. The assembly of ORF4 and PilE proteins in filaments makes a two-hybrid assay between them inappropriate. To visualise the possible interaction between a MDAΦ and a pilus, we first performed transmission electron microscopy (TEM) on whole bacteria incubated with phages. To prevent phage internalisation and to facilitate observation of the phage-pili interaction, we used a strain mutated for pili retraction and expressing the PilE$_{SD}$ variant. The ZΔMDAΔpilT pilE$_{SD}$ strain was thus incubated with purified bacteriophages, washed to remove free bacteriophages and processed for immunogold against the major coat protein ORF4 and TEM (Fig. 4a). Labelled phages were indeed found along the entire length of the pili bundles. Labelling of phage-free bacteria showed no staining (Fig. 4a).

We next assessed MDAΦ binding to pili by quantifying phage precipitation by bacteria expressing either the PilE$_{SA}$ or PilE$_{SD}$ variant. As above, we used strains mutated for pili retraction. We incubated purified MDA$_{(orf6-aadA1)}$Φ with the strains ZΔMDAΔpilT pilE$_{SA}$ or ZΔMDAΔpilT pilE$_{SD}$ and a pili-deficient strain (ZΔMDA ΔpilE) as a negative control (Fig. 4b, c). Bacteria were then washed and immobilised on glass slides before immunostaining with anti-ORF4 antibodies. The aera of ORF4 labelling was analysed using ImageJ and normalised to that of bacteria stained with DAPI (Fig. 4b). As expected, phages did not bind to the non-piliated ZΔMDA ΔpilE strain. The remaining signal for the non-piliated ZΔMDA ΔpilE strain likely corresponds to background and non-specific labelling. However, there is a significant difference in phage precipitation depending on the pilE sequence. MDA phages bound to the PilE$_{SA}$ variant at reduced levels, whereas they bound to the strain specifically expressing the PilE$_{SD}$ variant (Fig. 4b, c).

To better visualise the phage-pili interaction, we have decided to perform high-resolution STED (Stimulated Emission Depletion) microscopy on precipitated bacteria. However, as the PilE$_{SD}$ variant was not recognised by our anti-PilE 20D9 antibody, we had to derive a new strain with phage binding and transduction properties whose pili were labelled by our antibody. To this end, we followed the same strategy as that used to obtain the PilE$_{SD}$ strain. We transduced the ZΔMDA strain expressing the PilE$_{SB}$ variant (Fig. 1e and Supplementary Fig. 1c) and selected transduced derivatives. We have selected one new pili variant, named PilE$_{SE}$, which was favourable for phage entry and recognised by the 20D9 antibody (Supplementary Fig. 4a, b). We then incubated purified MDA$_{(orf6::aph3')}$Φ with the strain ZΔMDA ΔpilT pilE$_{SE}$. The bacteria were washed and immobilised on glass slides before immunostaining with anti-ORF4 and anti-pili 20D9 antibodies. The slides were then processed for high-resolution STED microscopy (Fig. 4d and Supplementary Fig. 4a). Pili staining is shown as dotted lines, which we had previously observed using high-resolution microscopy with the anti-pili 20D9 antibody[43,48], and most of the phage staining appears to be associated with pili along their entire length. We also quantified phage retention by ΔpilT strains expressing either the PilE$_{SB}$ or PilE$_{SE}$ variant and the amount of pili around the strains as in

Fig. 4b (Supplementary Fig. 4d, e). The phage retention is the ratio of phage surface labelling to bacterial surface labelling after phage precipitation (see materials and method). As expected, the strain expressing the PilE$_{SE}$ variant appeared to retain significantly more phage than the variant expressing PilE$_{SB}$ ($p = 0.0006$). Interestingly, the ZΔMDA ΔpilT pilE$_{SB}$ strain produces more pili than the ZΔMDA ΔpilT pilE$_{SE}$ strain, but the amount of pili around these two variants is not significantly different ($p = 0.5162$). Finally, to confirm the absence of a role for ORF6 in the interaction between the phage capsid and pili, we used a phage preparation produced from a strain deleted for orf6[17] (MDA$_{(orf6::aph3')}$Φ). We incubated the purified MDA$_{(orf6::aph3')}$Φ preparation with the strain ZΔMDAΔpilT pilE$_{SD}$ as above and used the pili-deficient strain (ZΔMDA ΔpilE) as a negative control (Supplementary Fig. 5). We observed no significant difference between the retention of the MDA$_{(orf6-aadA1)}$Φ and the MDA$_{(orf6::aph3')}$Φ on pili. This confirms that ORF6 does not play a role in the primary interaction between the phage and the pili. Taken together, these results suggest that, in contrast to what has been shown for other filamentous phages, MDAΦ preferentially interacts with pili along their fibres rather than at their tips, an association that strongly correlates with the transduction rate.

## MDAΦ transduction is associated with positively charged pili

The differences between the PilE$_{SA}$ and PilE$_{SD}$ variants are essentially differences in amino acids in the D-region, with amino acids having different charges, resulting in a change in the isoelectric point of the pilin (Fig. 5a and Supplementary Fig. 2b). Interestingly, the coat of MDAΦ, consisting of the major coat protein ORF4, has recently been modelled by cryo-EM[50]. This model showed that the MDAΦ surface was predominantly negatively charged. We therefore reasoned that the interaction between MDAΦ and pili might be facilitated by opposite charges on the negatively charged MDAΦ coat and positively charged pili such as those displaying the PilE$_{SD}$ variant. To test this hypothesis, we first selected four clones expressing different PilE variants from the ZΔMDA strain. These variants were selected on the basis of amino acid charge and were named PilE$_{Z1}$, PilE$_{Z2}$, PilE$_{Z3}$, and PilE$_{Z4}$, with PilE$_{Z1}$ being the most positively charged and PilE$_{Z4}$ the most negatively charged (Fig. 5a and Supplementary Fig. 2b). We then modelled in silico the pilus fibre of these strains and the structure of PilE$_{SA}$ and PilE$_{SD}$ using Alphafold and the cryo-EM reconstruction of the N. meningitidis type IV pilus (5KUA; https://www.rcsb.org/structure/5kua). The Fig. 5a shows the difference in charged amino acids exposed for the six PilE variants studied. PilE$_{SD}$ has a majority of positively charged amino acids on its surface, while PilE$_{SA}$ has a majority of negatively charged amino acids, with the other four strains falling between these two extremes. We then assessed the frequency of MDA$_{(orf6-aadA1)}$Φ transduction in these ZΔMDA clones that predominantly express pilE$_{Z1}$, pilE$_{Z2}$, pilE$_{Z3}$, or pilE$_{Z4}$ (Fig. 5b). We observed a correlation between pili charge and phage transduction: bacteria with positively charged pili were better transduced by phages than those expressing negatively charged pili.

We further examined the differences between the amino acid sequences of PilE$_{SA}$ and PilE$_{SD}$, focusing on the charged amino acids (Supplementary Fig. 2b). We decided to focus on position 70-71 and the D-region. We therefore constructed a ZΔMDA mutS$^+$ΔrecA pilE$_{SA}$ mutant in which we exchanged A$_{70}$D$_{71}$ with S$_{70}$K$_{71}$ of PilE$_{SD}$ and/or the D-region with that of PilE$_{SD}$, and vice versa between PilE$_{SD}$ and PilE$_{SA}$ (Fig. 5c). The structure of these mutants was then modelled using Alphafold to visualise charged amino acids (Supplementary Fig. 6a, b). We then assessed the MDA$_{(orf6-aadA1)}$Φ transduction rate in all these mutants (Fig. 5d). The exchange between the D-regions dramatically increased (PilE$_{SA}$ to PilE$_{SD}$) or decreased (PilE$_{SD}$ to PilE$_{SA}$) the transduction rate, highlighting the role of this region. Modifications at positions 70-71 had little effect on the pilin-exposed charges, and therefore had little effect on the transduction rate. These data support

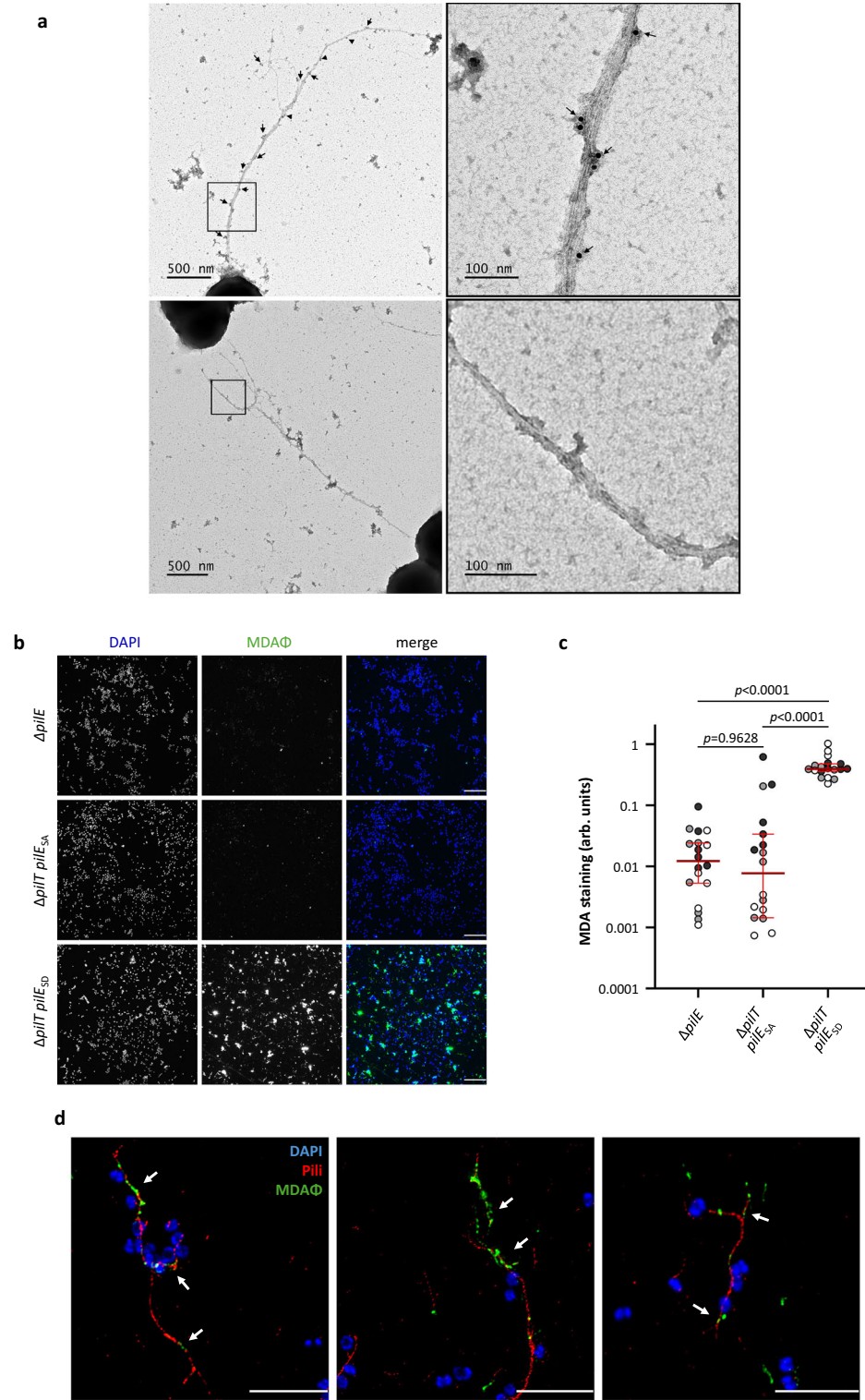

**Fig. 4 | Interaction between MDA phages and type IV pili. a** Representative immunogold labelling of strain ZΔ*MDA ΔpilT pilE_SD* incubated with MDA_(orf6-aadAI) phage (top) and a control without phage (bottom). Phages were labelled with the anti-MDA_ORF4_N-ter antibody coupled to 10 nm diameter gold particles. We did not observe any labelling of the type IV pili bundle by the antibody. The right images are enlargements of the framed area in the left images. Representative images from the observation of two independent experiments (*n* = 2). **b, c** MDA staining on strains expressing different PilE variants was estimated by immunofluorescence as the ratio of phage surface labelling to bacterial surface labelling after phage pre-cipitation (arb. units: arbitrary unit) (**b**) Statistical analysis was performed using a

Brown–Forsythe and Welch ANOVA test (two-sided) with Tamhane T2 correction, and data were expressed as median with 95% CI. Each dot represents one image and images were taken from three different experiments (*n* = 3), with dots coloured black, grey and white, respectively. Source data are provided as a Source Data file. Representative images used for the quantification are shown in b. Bar = 20 μm. **d** High-resolution STED microscopy was performed on MDA_(orf6-aadAI) phages on a preparation of the strain ZΔ*MDA ΔpilT pilE_SE* (see methods section). Phages and pili are labelled with anti-MDA_ORF4_N-ter antibody (anti-phage, green) and anti-PilE 20D9 antibody (anti-pili, red). Bacteria are stained with DAPI. Representative image from several observations of one experiment (*n* = 1). Bar = 5 μm.

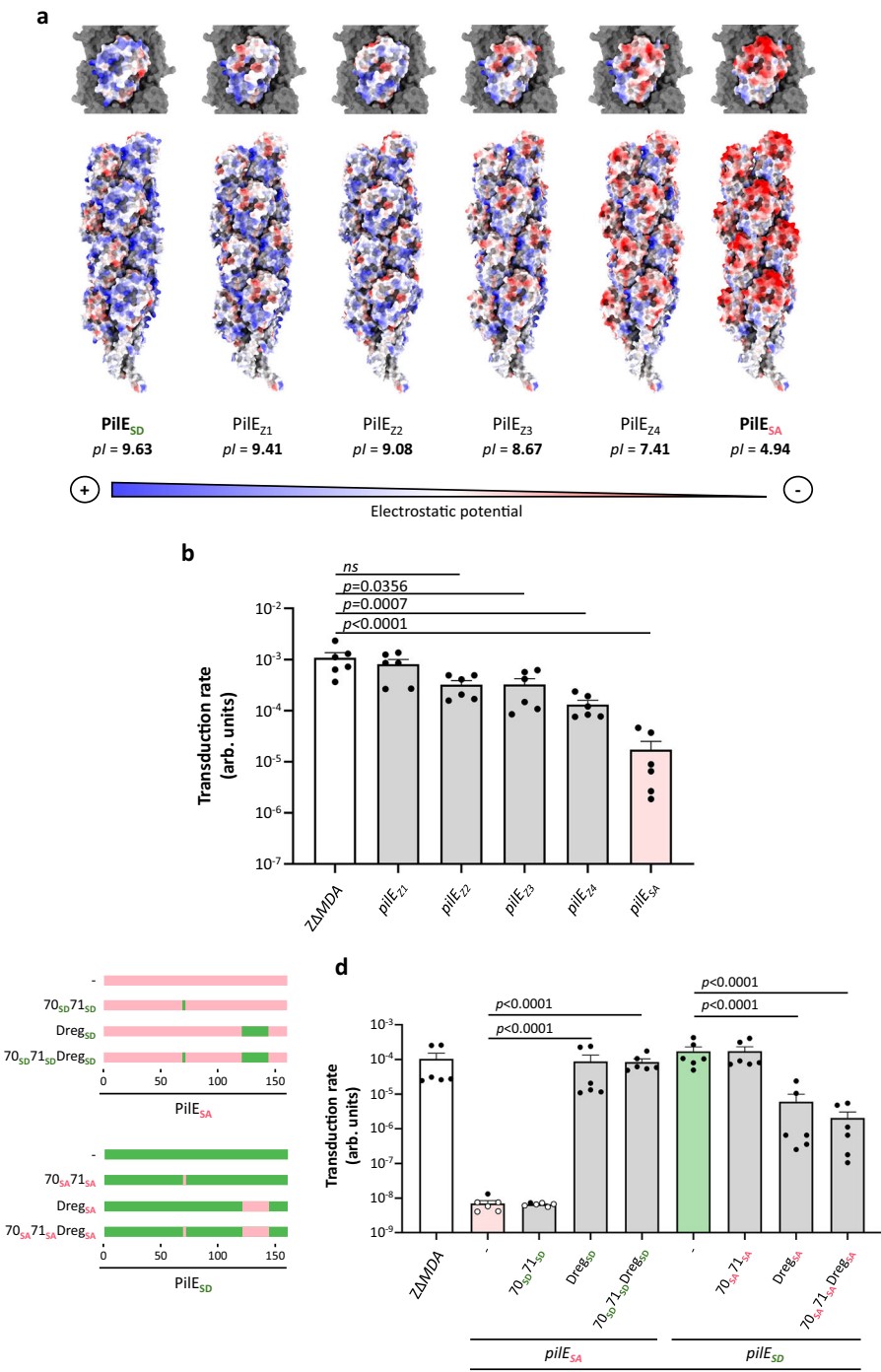

**Fig. 5 | Phage transduction depends on the charges exposed at the surface of PilE. a** Reconstruction of type IV pilus fibres for six different PilE variants from the ZΔMDA strain (this strain has a mix of *pilE* sequences), from the most positively charged surface (on the left) to the most negatively charged surface (on the right). The predicted structures are based on the biological assembly 5KUA and the predicted structure of the PilE subunits obtained using Alphafold-3. Structures were coloured using the coulombic electrostatic potential command of ChimeraX, with the default colour ranging from red for negative potential through white to blue for positive potential. pI (isoelectric point) was determined using the iep tool on Galaxy version 5.0.0.1 with initial parameters. **b** Transduction rate of ZΔMDA strain and its four derived clones (arb. units: arbitrary unit). The transduction rate of *pilE*$_{SA}$ has been reused from Fig. 1e to facilitate understanding of the figure. Experiments were performed three times in duplicate (*n* = 3). Statistical analyses were performed using the ordinary ANOVA test with Dunnett's correction, and data were expressed as mean ± SEM. Source data are provided as a Source Data file. **c** Amino acid sequence alignment scheme for strains expressing the PilE$_{SA}$ and PilE$_{SD}$ variants and their derivative mutants in which amino acid 70-71 and/or the D-region (Dreg) were swapped. The PilE$_{SA}$ sequence is shown in red and the PilE$_{SD}$ sequence is shown in green. **d** Transduction rate of strains ZΔMDA and its derivatives ZΔMDA *mutS*+ ΔrecA expressing PilE$_{SA}$ or PilE$_{SD}$ variants and their mutants. White dots correspond to CFU results below the detection threshold. Experiments were performed three times in duplicate (*n* = 3). Statistical analyses were performed using the Ordinary one-way ANOVA test with Bonferroni correction, and data were expressed as mean ± SEM. Source data are provided as a Source Data file.

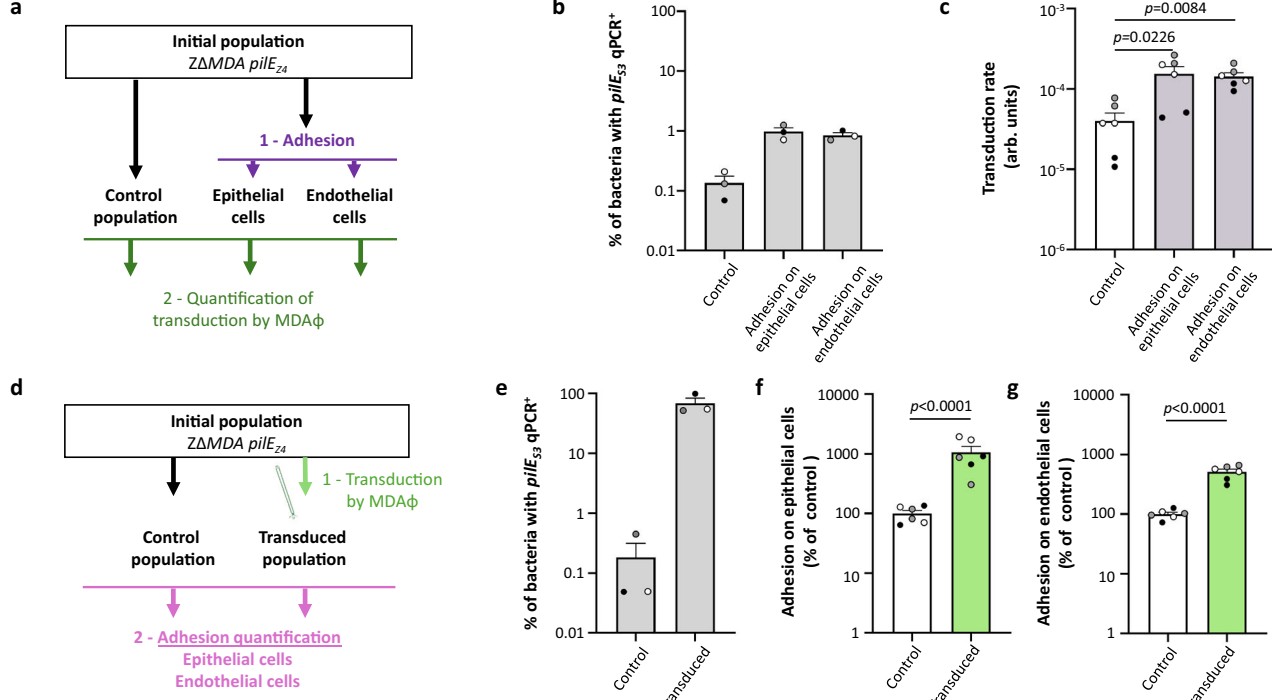

**Fig. 6 | Relationship between adhesion and phage transduction. a** Schematic representation of the methodology used in (**b**, **c**). An initial population of the ZΔ*MDA* expressing the PilE$_{Z4}$ variant was incubated in cell culture media or in the presence of epithelial or endothelial cells. In the former case, bacteria were collected (control). In the latter two cases, only adherent bacteria were collected. **b** The proportion of bacteria expressing a PilE$_{S3}$ variant was quantified by qPCR in the three populations. **c** The three resulting populations were transduced with MDA$_{(orf6-aadA1)}$Φ and the transduction rates were determined (arb. units: arbitrary unit). **d** Schematic representation of the methodology used in (**e–g**). An initial population of the ZΔ*MDA* expressing the PilE$_{Z4}$ variant was incubated in transduction media and collected (control) or transduced with MDA$_{(orf6-aadA1)}$Φ and

selected on spectinomycin (transduced). **e** Determination of the proportion of bacteria expressing a PilE$_{S3}$ variant in the two populations by quantification by qPCR. Comparison of the adhesion of the two populations on epithelial cells (**f**) and endothelial cells (**g**). Data were expressed as a percentage of the control population. **b**, **e** qPCR was performed three times independently ($n = 3$). **c**, **f**, **g** Experiments were performed three times in duplicate independently ($n = 3$). Dots coloured black, grey and white in **b**, **c** and **e–g** represent the results from each experiment, respectively. Statistical analyses were performed using a Brown–Forsythe and Welch ANOVA test (two-sided) with Dunnett correction for **c**, an unpaired *t*-test (two-sided) for **f**, **g**, and data were expressed as mean ± SEM. Source data are provided as a Source Data file.

our hypothesis that the interaction between MDAΦ and type IV pili benefits from a positively charged pilus.

### Adhesion and transduction both determine the selection of the same *pilS* cassette

Both MDAΦ and pili are associated with the invasiveness of *N. meningitidis*. We therefore sought to determine whether adhesion to human cells could select for a PilE variant that favours transduction by MDAΦ or whether strains transduced by MDAΦ could select for a more adhesive PilE variant. We have previously shown that transduction of MDA$_{(orf6-aadA1)}$Φ in a ZΔ*MDA pilE$_{SA}$* strain, after spectinomycin selection, resulted in a population expressing predominantly the PilE$_{SD}$ variant (Fig. 2e), which appeared to correspond to an antigenic variation achieved between the *pilE* gene and the *pilS3* cassette (Fig. 2d and Supplementary Fig. 2a, b). To assess the proportion of *pilS3*/*pilE$_{SD}$* variant in an adhesive or transduced population, we designed a qPCR assay with one primer specific to the *pilS3* sequence and one primer specific to the constant region of *pilE* (Supplementary Fig. 2b). A second pair of primers targeting *pilT* was used as a control. We validated this method on DNA samples from two ZΔ*MDA* Δ*recA* strains (which avoid antigenic variation) expressing PilE$_{SD}$ or PilE$_{Z4}$ corresponding to an antigenic variation achieved between the *pilE* gene and the *pilS8* cassette (Supplementary Fig. 2a, b). As expected, we obtained a proportion of *pilE$_{S3}$* positive bacteria close to 100% for the ZΔ*MDA* Δ*recA pilE$_{SD}$* strain, and a proportion of *pilE$_{S3}$* positive bacteria of 0.01% for the ZΔ*MDA* Δ*recA pilE$_{Z4}$* strain

(Supplementary Fig. 7a). We then examined the proportion of bacteria carrying a recombination of the *pilE* gene with the *pilS3* locus in the ZΔ*MDA pilE$_{Z4}$* strain. Only 0.1% of the population was *pilE$_{S3}$* positive (Supplementary Fig. 7b). This strain was further used to assess the proportion of bacteria expressing the PilE$_{S3}$ variant after adhesion to endothelial or epithelial cells and then the subsequent transduction by MDA$_{(orf6-aadA1)}$Φ. In a first experiment (Fig. 6a), we collected ZΔ*MDA pilE$_{Z4}$* bacteria adherent to FaDu epithelial cells or EA.Hy926 endothelial cells or after incubation in DMEM, and we quantified both the proportion of bacteria expressing PilE$_{S3}$ and the frequency of transduction. The proportion of bacteria expressing PilE$_{S3}$ increased after adhesion to epithelial and endothelial cells (Fig. 6b) from 0.13% of the population to almost 1% and 0.85% respectively. Similarly, adhesion to epithelial and endothelial cells increased the transduction rate by almost fourfold (Fig. 6c, $p = 0.0226$ and $p = 0.0084$, respectively). In a second experiment (Fig. 6d), we transduced the ZΔ*MDA pilE$_{Z4}$* strain with MDA$_{(orf6-aadA1)}$Φ and quantified the adhesion of the resulting population. The initial ZΔ*MDA pilE$_{Z4}$* strain was incubated in transduction media without phage and kept as a control. The population infected by the phage and selected on spectinomycin became predominantly *pilE$_{S3}$* positive (68% of the total population), whereas it remained at 0.18% in the control population (Fig. 6e). After transduction, the spectinomycin-resistant population was 10.6 times more adhesive to epithelial cells and 4.7 times more adhesive to endothelial cells than the initial population (Fig. 6f, g, $p < 0.0001$ and $p < 0.0001$ respectively). Therefore, there is a strong

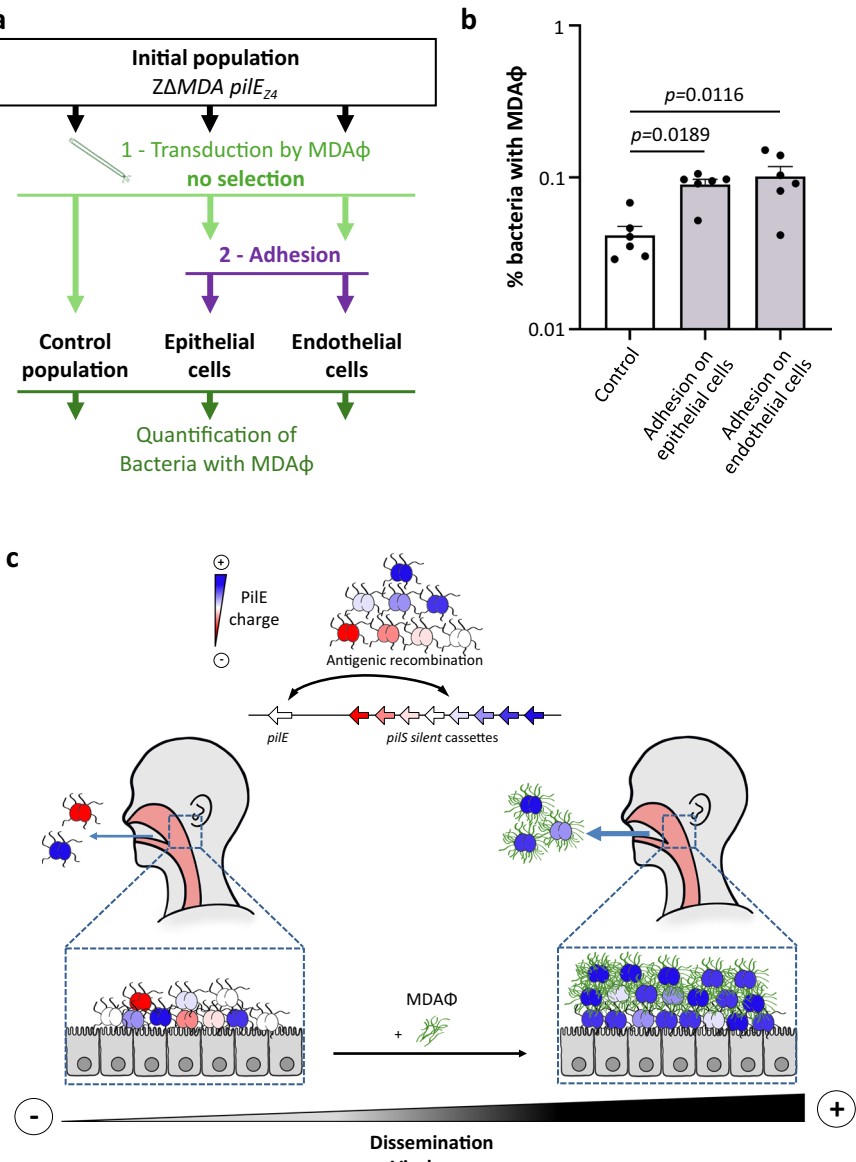

**Fig. 7 | Phage enrichment in adherent bacteria. a** Schematic representation of the methodology used in (**b**). An initial population of the ZΔ*MDA* expressing the PilE$_{Z4}$ was transduced with MDA$_{(orf6-aadA1)}$Φ and recovered without selection to have both transduced and non-transduced bacteria in the same population. The population was then incubated in cell culture media and collected or incubated in the presence of epithelial or endothelial cells and only adherent bacteria were collected. The three populations were then plated on agar containing spectinomycin to quantify the percentage of bacteria infected by MDA$_{(orf6-aadA1)}$Φ. **b** The proportion of bacteria with MDA$_{(orf6-aadA1)}$ prophage was determined for the three populations. **b** Experiments were performed three times in duplicate ($n = 3$). Statistical analyses were performed using a Kruskal–Wallis test with Dunn's correction for b, and data were expressed as mean ± SEM. Source data are provided as a Source Data file.

**c** Schematic representation of our proposed model of the amplification loop between MDAΦ infection, biofilm formation and enrichment in adhesive PilE variants. The top panel shows the expression of eight possible PilE variants following antigenic variation and coloured according to their PilE charge. Bottom left: a human carrying strains of *N. meningitidis* with different pili variants and not infected by the MDA phage will spread any PilE expressing bacteria. Bottom right: when an isolate expressing the MDA phage is present in the population, the phage infects bacteria expressing the most adherent PilE variants, which are able to promote themselves in the niche and produce phages. These phage-producing bacteria will form a biofilm at the top of the population[11] and will be preferentially disseminated by the host.

link between transduction and adhesion by selection of a specific *pilE* variant.

### Enrichment of phage-infected bacteria during adhesion

Finally, we asked whether adhesion to human cells would be sufficient to increase the proportion of phage-containing bacteria in a given bacterial population. To answer this question, we brought a ZΔ*MDA pilE*$_{Z4}$ population into contact with purified MDA$_{(orf6-aadA1)}$Φ without further antibiotic selection. Bacteria were then incubated on epithelial and endothelial cells or incubated in DMEM as a control.

After 3 h, the cells were washed, adhesive bacteria were recovered and the proportion of phage-containing bacteria was assessed by serial dilution on agar plates containing spectinomycin and CFU counting (Fig. 7a). As a control, we confirmed that the growth and adhesion of the two bacterial populations - phage infected or not - were similar (Supplementary Fig. 7c). Bacterial adhesion to epithelial or endothelial cells was sufficient to increase the MDA-carrying population by 2.1 and 2.4 times, respectively, compared to the total population (Fig. 7b, $p = 0.0189$ and $p = 0.0116$ respectively). This result highlights the mutual benefit of the MDAΦ-meningococcus

interaction in promoting both phage propagation and *N. meningitidis* colonisation (Fig. 7c).

## Discussion

In this study, we have dissected the molecular interaction between MDAΦ and type IV pili, an interaction that is likely to be intimately involved in shaping the population of PilE variants and thus influencing meningococcal colonisation. We characterised an association between MDAΦ, type IV pili and host colonisation, providing new insights into the role of MDAΦ in the interaction of meningococci with their host and suggesting an unexpected link between the life cycle of this filamentous phage and antigenic variation.

First, we studied phage transduction in type IV pili machinery mutants, and we demonstrated the absolute necessity of retractable pili. Among the piliated mutants of the type IV pili machinery, only the *pilT* ATPase mutant was not infected by phages. The *pilT2* and *pilU* genes, which encode two other proteins known to be members of the ATPase family of the type II/IV secretion system, were not required for phage infection, which seems consistent with the absence of complementation of a *pilT* mutant by *pilT2* or *pilU*, as previously shown by Brown et al.[30] for transformation with exogenous DNA. Non-piliated mutants, such as those deleted in *pilH, I, J*, or *K* or both *pilC1* and *pilC2*, were weakly infected by phages, but more so than a *pilE* mutant, although it is surprising that pili fibres can be extruded from the PilQ pore without being capped by the tip complex. Another surprising result is the level of transduction and transformation observed in the Δ*tfpC* mutant. Although TfpC has been described as playing a role in maintaining the extended state of pili, it has not been identified as a factor in pili expression[51]. Furthermore, Muir and collaborators found that this protein is involved in transformation efficiency[52], which was not observed in our study. The Δ*tfpC* mutant is as transformable and transductible as the wild type, but appears to have slightly fewer pili. It does not appear to be involved in transduction by MDAΦ. This phenotype was observed for insertion and deletion mutant in two different strains. We then investigated in detail the possibility that PilC proteins are involved in MDAΦ binding. The two-hybrid assay failed to demonstrate a possible interaction between the PilC N-terminal domain and ORF6. Besides, our experiments with the PilC deletion mutants and that of the extremely weak phage transduction observed in the PilE$_{SA}$ variant strongly suggest that PilE is the only pilin required for phage transduction, and that the pilus tip proteins PilC, H, I, J, K are probably not necessary. Although unlikely, we may not exclude that the phage binds with the very last part of the C-terminal domain of PilC. Similarly, the interaction with PilQ or the involvement of proteins whose absence results in a non-piliated phenotype (such as PilD, PilF, PilP) cannot be excluded. In addition, direct observation by electron microscopy and high-resolution microscopy revealed an interaction between MDAΦ and type IV pili along the pili fibre. To our knowledge, this work is the first demonstration, at the molecular level, of an interaction between the capsid of a filamentous phage and the lateral fibre of type IV pili rather than between the tip of the phage and the tip of the pilus.

We have previously published that the ORF6, located at one end of the phage, should be the adsorption protein required for phage entry across the bacterial membranes[17]. This conclusion was based on the absence of transduction by MDAΦ mutated for *orf6* and on the results obtained for the adsorption proteins of phage Ff and CTX, which recognise pili ends[15,53]. The Ff and CTX phage adsorption proteins are also involved in the crossing of the inner membrane via an interaction with the TolQRA system. Therefore, although we demonstrated that ORF6 is not involved in the binding to pili, this does not rule out a role for ORF6 in inner membrane crossing. This step needs to be further investigated in the future.

In meningococci, antigenic variation of *pilE* plays an important role in cell adhesion to the host and that of escape from the immune response[34,35]. Antigenic variation of *pilE* with one of the silent *pilS* loci allows large modifications of the amino acids exposed on the pili surface, leading to changes in charge and hence isoelectric point of pilin in only a subset of the population. Unlike the *N. meningitidis* laboratory strain 8013, strain Z5463 has not been selected for adhesion to host cells[34] and is expected to be able to express a diversity of *pilE* sequences (due to antigenic variation and the hypermutator phenotype of the strain[29]). This allows us to observe the preferential infection of phages in *N. meningitidis* expressing PilE variants associated with high PI, although this variant is expressed by a minority of bacteria. One hypothesis that explains the association of the MDA phages with these PilE variants is electrostatic attraction between the positively charged pilin proteins and the bacteriophages whose capsids are mainly negatively charged, as recently shown by Böhning et al.[50], using cryoEM, although we cannot exclude an interaction between the phage capsid and specific amino acids on PilE.

We found a very strong relationship between effective adhesion and effective transduction, with the PilE variant with the highest isoelectric point being the most adhesive and most transducible variant. Therefore, the presence of one MDA phage in a *N. meningitidis* genome is likely to be associated with the presence of a highly adhesive PilE variant in the population. *N. meningitidis* genomes contain one to four different MDA prophages, corresponding to multiple infections[28] and our works and that of others strongly suggest that infected bacteria are competent for multiple infection by MDA phages[10,17].

Finally, the experiment presented in Fig. 7a more closely resembles the natural life cycle of phage and *Neisseria*, i.e. without direct selection for the presence of the MDAΦ. In this latter experiment, the adhesion to human cells was sufficient to increase the population of MDAΦ-positive bacteria. Considering our results, the infectious cycle of this phage and its role in biofilm formation, we propose a model in which an amplification loop - between MDAΦ infection, biofilm formation and enrichment in adhesive PilE variants - is likely to favour the emergence and maintenance of MDAΦ-positive clones in the most adhesive population, which is also potentially the most virulent (Fig. 7c).

In conclusion, we have described a unique mode of interaction between a filamentous phage and the type IV pili of its host revealing a new type of mutual benefit between a phage and its host.

## Methods

### Bacterial strains and culture media

The *N. meningitidis* strains used in this study are listed in Supplementary Data 1 and are derived from Z5463, a naturally transformable serogroup A strain isolated from the throat of a patient with meningitis in the Gambia in 1983[54]. *N. meningitidis* mutants used in this work are listed in the Supplementary Data 1. It should be noted that the ZΔ*MDA* strain was constructed from the wild-type Z5463 strain. The initial mutant strain was clonal (one colony was selected), but underwent several passages to eliminate the circular cytoplasmic form of MDAΦ. Considering that the frequency of antigenic variation in *N. meningitidis* is 1 to $2 \times 10^{-3}$, the resulting strain expresses several different PilE sequences. *E. coli* strain DH5α was used for DNA cloning and plasmid propagation. *E. coli* Oxi-Blue[55] was used for two-hybrid assay. *E. coli* Oxi-Blue, derived from the Oxi-BTH strain, facilitates the correct folding of proteins formed by disulphide bonds within its cytoplasm.

The *N. meningitidis* strains were grown at 37 °C in 5% $CO_2$ on Brain Heart Infusion Agar (BHI Agar, Condalab) plate containing 5% of heat-inactivated horse serum (Gibco), or in gonococcal base (GCB) agar (Difco) plates containing 12 μM $FeSO_4$ and Kellogg's supplements[56] or in BHI liquid medium (Condalab) or in GC-liquid medium [1.5% proteose peptone (Difco); 0.4% $K_2HPO_4$, 0.1% $KH_2PO_4$, 0.1% NaCl], sometimes with 12 μM $FeSO_4$ and Kellogg's supplements. *E. coli* was grown at 30 °C or 37 °C on Lysogeny Broth (LB) agar or in LB liquid medium. For antibiotic selection[3] of *E. coli* strains, kanamycin (Km) was used at a concentration

of 50 µg/ml, erythromycin (Em) at 150 µg/ml, and ticarcillin (Tic) at 100 µg/ml. To select strains derived from *N. meningitidis*, Km was used at a concentration of 200 µg/ml, spectinomycin (Sp) at 75 µg/ml, Em at 3 µg/ml, chloramphenicol (Cm) at 5 µg/ml, nalidixic acid (Nal) at 1 µg/ml, and tetracyclin (Tet) at 2.5 µg/ml.

## Cell lines and culture conditions
The pharynx carcinoma-derived epithelial cell line (FaDu; ATCC #HTB-43) and the human umbilical vein cell line (EA.hy926; ATCC #CRL-2922) were obtained from the American Type Culture Collection (ATCC). The FaDu and EA.hy926 cell lines were grown in Dulbecco's Modified Eagle Medium (DMEM; Gibco) supplemented with 10% decomplemented foetal calf serum (FCS; Gibco). Cells were grown at 37 °C in a humidified incubator under 5% $CO_2$.

## Phage preparation, quantification, and transduction assay
**Phage preparation.** Phage preparation was performed as previously described using two different strains: ZS2 strain is a derivative of strain Z5463 to which a Sp resistance gene (*aadA1*) has been added in the MDA prophage (MDA$_{(orf6-aadA1)}$Φ), just after *orf6*, without promoter, and Z5463 *orf6::aph3'* strain, previously described[17], derivative from Z5463 with prophage gene *orf6* replaced by a kanamycin resistance cassette without promoter (MDA$_{(orf6::aph3')}$Φ). Bacteria were pelleted from 200 ml of an overnight culture in BHI liquid medium containing Sp at 200 µg/ml (for ZS2) or Km at 300 µg/ml (for Z5463 *orf6::aph3'*). After filtration at 0.45 µm, the supernatant was treated with DNase I (25 µg/ml) for 1 h at 20 °C. Phage was precipitated by the addition of 0.5 M NaCl and 10% polyethylene glycol 6000 and incubated overnight at 4 °C. The phage was then pelleted by centrifugation at $10,000 \times g$ for 30 min and resuspended in PBS 1X (with $CaCl_2$ and $MgCl_2$; Gibco). The phage suspension was centrifuged at $200 \times g$ for 10 min, and the supernatant was collected to remove insoluble particles. It was then filtered through a 0.45 µm filter to prevent contamination. The purified phage preparation was used directly for immunogold labelling, immunofluorescence, or bacterial transduction assays.

## qPCR quantification of phage concentration
Phage concentration was quantified by qPCR using two primer pairs: one targeting the phage genome (ORF6-F/R for MDA$_{(orf6-aadA1)}$Φ or ORF4-F/R for MDA$_{(orf6::aph3')}$Φ), and another targeting the Glucose-6-phosphate dehydrogenase (G6PD-F/R) to assess total bacterial chromosome number. qPCR was performed with Luna Universal qPCR Master Mix (NEB) on a QuantStudio 1 (Applied Biosystems), run in triplicates. Cycling conditions were 95 °C for 1 min, followed by 40 cycles of 95 °C for 15 s and 60 °C for 30 s. A standard curve was generated using genomic DNA from strain Z5463Δ*orf1*, assumed to contain a single phage copy per genome. The resulting standard equations enabled quantification of circular phage DNA, the relative amount of phage is obtained by Number$_{ORF6/ORF4}$ – Number$_{G6PD}$. After quantification, the average phage quantity is estimated to be $2 \times 10^7$ phage/µl.

## Transduction assay
Recipient strains were grown overnight on agar plates and resuspended at an $OD_{600}$ of 1 in transduction liquid medium [containing GC-liquid medium, 12 µM $FeSO_4$, Kellogg's supplements, 2.5 mM $MgCl_2$, 2.5 mM $MgSO_4$, and 25 µg/ml DNase I]. Two hundred microliters of the bacterial suspension (corresponding to $2 \times 10^8$ bacteria) were aliquoted into a 24-well plate and mixed with 30 µl of the MDA$_{(orf6-aadA1)}$ phage preparation. This corresponds approximately to a bacteria:phage ratio of 1:3. The mix was incubated for 15 min at 37 °C with shaking and 15 min without shaking, and 800 µl of transduction liquid medium was added to each well. Plates were incubated for a further 2 h at 37 °C without shaking. If necessary, bacteria were plated on agar plates with or without Sp and stored the next day at −80 °C in BHI/15% glycerol.

To determine the transduction rates, bacteria were diluted and plated on agar plates with or without Sp before counting of CFU the next day. The transduction rate was calculated by dividing the number of CFU obtained on Sp-containing agar plates by that on antibiotic-free agar plates. The efficacy of the phage preparation was verified by transducing 30 µl of the phage preparation into the ZΔ*MDA* strain and was validated if the transduction rate was greater than $10^{-3}$. If no bacteria were obtained at the least diluted point on the agar plates, a CFU of 0.9 was arbitrarily assigned and the point was marked as below the detection limit. A detection threshold corresponding to the frequency of spontaneous resistance to spectinomycin was experimentally calculated using the ZΔ*MDA* strain ($3.2 \times 10^{-8} \pm 4 \times 10^{-8}$ – SD; $n = 3$ experiments in duplicate).

## Transformation of *N. meningitidis*
Strains of interest were grown overnight on BHI agar plates and resuspended to $OD_{600} = 1$ in liquid transformation medium [containing GC-liquid medium, 12 µM $FeSO_4$, Kellogg's supplements, $MgCl_2$ at 2.5 mM, and $MgSO_4$ at 2.5 mM]. Five µl of PCR product or 150 ng of plasmid or three µl of genomic DNA were added to 200 µl of this bacterial suspension and incubated for 30 min at 37 °C, 5% $CO_2$ with shaking. Bacteria were then grown for 2 h after the addition of 800 µl of liquid transformation medium. Transformants were selected on BHI agar plates containing corresponding the appropriate selective antibiotic.

To determine the transformation frequency, bacteria were resuspended to $OD_{600} = 0.1$ in step 1 transformation media [containing GC-liquid medium, $MgSO_4$ at 5 mM]. One thousand ng of genomic DNA carrying a Nal-resistant cassette was added to 200 µl of this bacterial suspension and incubated for 30 min at 37 °C, 5% $CO_2$ with shaking. Bacteria were then grown without agitation for 2 h after the addition of 800 µl of step 2 transformation liquid medium [containing 12 µM $FeSO_4$ and Kellogg's supplements except L-Glutamine]. This experiment has been described previously[30]. After incubation, bacteria were diluted and plated on agar plates with and without Nal before CFU counts the next day. The transformation frequency was calculated by dividing the number of CFU obtained on Nal-containing agar plates by the number of CFU obtained on antibiotic-free agar plates. If no bacteria were obtained at the least diluted point on the agar plates, a CFU of 0.9 was arbitrarily assigned, and the point was marked as below the detection limit.

## PCR amplification, plasmid construction and mutant generation
**PCR amplification and purification.** Sequence amplifications for construction and sequencing were conducted using Q5 High Fidelity DNA polymerase (New England Biolabs) and the oligonucleotides listed in Supplementary Data 2. All oligonucleotides were purchased from Integrated DNA Technologies (eu.idtdna.com). After amplification, PCR products amplified from plasmids were incubated for 1 h at 37 °C with *Dpn*I enzyme (NEB) to digest the methylated plasmid backbones. PCR products were purified using Monarch DNA Gel Extraction kit (New England Biolabs) or Monarch PCR & DNA Cleanup Kit (New England Biolabs). All mutants were verified by PCR amplification using OneTaq DNA Polymerase (New England Biolabs) on bacterial genomic DNA extracted using the Wizard Genomic DNA purification kit (Promega) according to the manufacturer's instructions.

## Plasmid construction and mutant generation
All plasmids used in this work are listed in the Supplementary Data 3.

*ZΔMDA pilE$_{SA}$ and ZΔMDA pilE$_{SB}$.* The *pilE$_{SA}$* and *pilE$_{SB}$* alleles fused to the Km resistance gene and the flanking regions (≈250 pb upstream and downstream) were amplified by PCR from the genomic DNA of strains NEM8013*pilE$_{SA}$* and *-pilE$_{SB}$* with primers 5'_pilE_Fw and 3'_pilE_Rv[34] and used for transformation and allelic exchange in

Z$\Delta$MDA. Bacteria were then isolated on Km, and *pilE* sequences were verified by Sanger sequencing. The same methodology was used to construct Z$\Delta$MDA *pilE*$_{SD}$ and Z$\Delta$MDA *pilE*$_{SE}$ with PCR amplification from the genomic DNA of transduced clonal strains obtained respectively from transduction of Z$\Delta$MDA *pilE*$_{SA}$ and Z$\Delta$MDA *pilE*$_{SB}$.

*Obtention of mutS$^+$ derivatives.* We took advantage of the plasmid containing the Z2491*mutS$^+$* allele with a Sp resistance cassette inserted after the gene, leaving a downstream intergenic sequence at the 3'end, allowing the allelic exchange to be performed (as described by Omer et al.[29]). The Sp resistance gene was swapped with a Cm resistance gene by PCR mutagenesis using appropriate primers (Supplementary Data 2). The Z$\Delta$MDA *mutS$^+$* strain was obtained by transformation of a vector containing the *mutS$^+$* allele with Cm resistance downstream of the gene and allelic exchange. Bacteria were then isolated on Cm and verified by PCR.

*Z$\Delta$MDA mutS$^+$ and Z$\Delta$MDA mutS$^+$ $\Delta$recA -pilE$_{SA}$ and -pilE$_{SD}$ and derived mutants.* The *pilE*$_{SA}$ and *pilE*$_{SD}$ alleles fused to the Km resistance gene and their flanking regions ($\approx$250 pb upstream and downstream) were cloned into the pMiniT 2.0 vector as above and using primers 5'_pilE_Fw and 3'_pilE_Rv. The resulting plasmids (namely pMiniT_*pilE*$_{SA}$-Km$^R$ and pMiniT_*pilE*$_{SD}$-Km$^R$) were sequenced and used as templates for site-directed PCR mutagenesis using the appropriate primers (Supplementary Data 2). PCR products corresponding to the *pilE* loci were obtained from wild-type and mutagenised pMiniT_*pilE*$_{SA}$-Km$^R$ or pMiniT_*pilE*$_{SD}$-Km$^R$, using primers 5'_pilE_Fw and 3'_pilE_Rv and were introduced into *N. meningitidis* strain Z$\Delta$MDA *mutS$^+$* by transformation and allelic exchange. Bacteria were then isolated on Km plates, and *pilE* sequences were verified by Sanger sequencing. To obtain Z$\Delta$MDA *mutS$^+$* $\Delta$recA -pilE$_{SA}$, -pilE$_{SD}$ or other derivatives variants of *pilE*, a genomic DNA from Z$\Delta$MDA $\Delta$recA was introduced into strains Z$\Delta$MDA *mutS* -pilE$_{SA}$, -pilE$_{SD}$ or other derivatives variants of *pilE* by transformation and allelic exchange. Bacteria were isolated on Tet plates, and *pilE* sequences were verified by Sanger sequencing and *recA* deletion was verified by PCR.

*Z$\Delta$MDA pilE$_{SB}$.* The *pilE*$_{SB}$ allele was cloned into pMiniT resulting in plasmid pMiniT_*pilE*$_{SB}$-Km$^R$. The Km resistance cassette fused to *pilE* gene was swapped with a Sp resistance gene by PCR-based mutagenesis using appropriate primers (Supplementary Data 2). The resulting *pilE*$_{SB}$ variant was introduced into strain Z$\Delta$MDA at the locus of *pilE* by transformation. Cloning and transformation strategies were the same as described for pMiniT_*pilE*$_{SA}$-Km$^R$.

*pNM99Ptet-pilC1 and -pilC2 and derived plasmids.* The *pilC1* and *pilC2* genes from strain Z5463 were cloned into the pMiniT 2.0 vector as described above. To avoid phase variation, the poly(G) sequences present in the two *pilC* signal peptide sequences were swapped with an amino acids identical alternative sequence using PCR mutagenesis with the primers C1.Z_polyG_Fw/Rv and C2.Z_polyG_Fw/Rv. The resulting plasmids, pMiniT-*pilC1*$_{fixed}$(Z5463) and pMiniT-*pilC2*$_{fixed}$(Z5463), were sequenced and used as a matrix for cloning of *pilC1* and *pilC2* into pNM99$_{Ptet}$[57], where gene expression is under the control of a *tet* promoter (primers pNM99_TET_Fw/Rv, pilC1_Z5463_pNM99_Fw/Rv, pilC2_Z5463_pNM99_Fw/Rv). Deletion mutants of *pilC1* were obtained by PCR mutagenesis using the appropriate primers directly on the pNM99$_{Ptet}$-*pilC1* plasmid. Mutations were verified by Sanger sequencing.

*Z$\Delta$MDA $\Delta$pilC1 $\Delta$pilC2 complemented for PilC1 and PilC2 and derived mutants.* Markerless null mutants were constructed as described by Wuckelt et al.[57]. Briefly, the gene of interest was first swapped by allelic exchange with the APG cassette carrying a Km resistance marker (aphA-3) and two counter-selectable marker genes (*pheS$^*$* and *galK*). The markerless null mutant was then obtained by removing the APG cassette by allelic exchange using a PCR product of the 5' and 3' regions flanking the gene of interest. Using this strategy, we first obtained a Z$\Delta$MDA $\Delta$pilC2 mutant and we inserted the complementation construct from the pNM99$_{Ptet}$-*pilC1* and -pilC2 plasmids (see above) by transformation and allelic exchange. Complemented

bacteria were selected on Em plates. We then obtained the $\Delta$pilC1 mutant using the same strategy as for the $\Delta$pilC2 mutant. The PilC expression, required for transformation, was allowed by the addition of anhydrotetracycline (20 ng/ml). The primers used are described in the Supplementary Data 2.

*Transposon mutants ($\Delta$pilC1, $\Delta$pilD, $\Delta$pilE, $\Delta$pilF, $\Delta$pilG, $\Delta$pilH, $\Delta$pilI, $\Delta$pilJ, $\Delta$pilK, $\Delta$pilQ, $\Delta$pilT, $\Delta$pilT2, $\Delta$pilU, $\Delta$pilV, $\Delta$pilW, $\Delta$pilX, $\Delta$pilZ, $\Delta$tsaP, $\Delta$pglA, $\Delta$pglC) and other deletion mutants ($\Delta$pilP, $\Delta$pilC2, $\Delta$NMA0415, $\Delta$tfpC, $\Delta$pptB, $\Delta$pilC1/C2, $\Delta$pilT, $\Delta$recA).* We took advantage of mutants originally generated in strain NEM8013 by in vitro transposition[58]. Regions of interest were amplified by PCR on the genomic DNA and then introduced into Z$\Delta$MDA by transformation of purified PCR products and allelic exchange. Bacteria were then isolated on Km plates, and mutants were verified by PCR. Other deletion mutants (except for the markerless mutants $\Delta$pilC1 and $\Delta$pilC2, see above) were obtained by inserting an antibiotic resistance cassette in place of the target gene (from START to STOP codons, except for *pilC2* where the last 537 nucleotides were retained) by allelic exchange after transformation into the strain Z$\Delta$MDA. The PCR product carrying the cassette flanked by regions upstream and downstream of the target gene was obtained by overlap PCR using primers listed in Supplementary Data 2. All transformants were selected on BHI agar plates containing a selective antibiotic. Mutations in the piliation machinery and pilus fiber genes were also introduced into strain Z$\Delta$MDA *pilE*$_{SB}$ by transformation with genomic DNA from the corresponding Z$\Delta$MDA mutant, followed by allelic exchange.

*Two-hybrid system.* The Bacterial Adenylate Cyclase Two-Hybrid System Kit (BACTH System Kit) from Euromedex was used. BACTH plasmids were designed to detect adenylate cyclase-fused protein expression. A 6His-tag was added to pKNT25/pKT25 and a FLAG-tag was added to pUT18/pUT18C. Divergent primers carrying sequences for the tag halves were designed to anneal just after the ATG start codon or just before the stop codon of the free extremity of the adenylate cyclase subunits. The PCR products form the complete tag upon plasmid ligation (same ligation method as for *pilE* mutagenesis). Then, to add inserts to the BACTH vector backbone, complementary primers were designed to overlap on 15–30 bp, and the two PCR products (insert and vector backbone) were purified and assembled using the NEBuilder HiFi DNA Assembly kit (NEB). The plasmids corresponding to *tolA3* in pUT18C, *gIII* in *pKT25*, *orf6* in pKNT25, pUT18, pKT25, pUT18C, *pilC1*$_{1-475}$ in pKNT25, pUT18, *pilC2*$_{2-492}$ in pKNT25, pUT18, *pilC1*$_{476-1021}$ in pKT25, pUT18C, *pilH*$_{30-213}$ in pKT25, pUT18C, *pilI*$_{35-189}$ in pKT25, pUT18C, *pilJ*$_{40-314}$ in pKT25, pUT18C, *pilK*$_{53-197}$ in pKT25, pUT18C are listed in Supplementary Data 3.

## Bacterial adenylate cyclase two-hybrid assay

Competent Oxi-Blue *E. coli* were co-transformed with 10 ng each of two recombinant plasmids encoding T18 and T25, respectively, Flag and 6His tagged. Transformed bacteria were plates on LB-agar plate supplemented with ampicillin and kanamycin. For each co-transformants, 3 clones were inoculated in duplicate (2×3 clones) into LB wells with Tic, Km, and IPTG 0.5 mM (isopropyl-$\beta$-D thiogalactopyranoside). Plates were incubated for 24 h at 37 °C with shaking, then 5 $\mu$l of each culture were plated in duplicate on LB agar plates supplemented with Tic, Km, IPTG 0.5 mM and X-gal 40 $\mu$g/ml (5-bromo-4-chloro-3-indolyl-$\beta$-D-galactopyranoside). Plates were incubated at 30 °C, and the colour of the colonies was assessed after 24 to 48 h.

## Adhesion to human cells

The 24-well plates were seeded with EA.Hy926 or FaDu cells and grown to confluence. Bacterial strains grown overnight on BHI agar plates were adjusted to OD$_{600}$ = 0.05 in DMEM/10% FCS and cultured for 2 h at 37 °C with agitation and 5% CO$_2$. The number of bacteria in the inoculum was estimated and cells were infected with $5 \times 10^6$ bacteria (Fig. 6a–g) or $1 \times 10^8$ bacteria (Fig. 7a, b). The exact number of CFU was

determined by serial dilution and enumeration of CFU. After 1 h (Fig. 6d) or 3 h (Figs. 6a and 7a and Supplementary Fig. 7) of contact, unbound bacteria were removed by six washes with 500 µl of PBS and the cells were detached by mechanical action in 500 µl DMEM/10% FCS. If necessary, bacteria were plated on BHI agar plates and stored the next day at −80 °C in BHI/15% glycerol. To determine the adhesion frequency, adherent bacteria were diluted and spread on agar plates before counting of CFU the next day. The adhesive frequency was obtained by dividing the number of CFU of adherent bacteria by the number of CFU of the inoculum.

### Bacterial preparation prior to immunolabeling for pilus quantification

In a 24-well plate, $2 \times 10^7$ bacteria were deposited onto coverslips precoated with D-polylysine the day before. The plate was centrifuged at $2500 \times g$ for 1 min and then incubated at 37 °C for 40 min. The medium was removed and the wells were washed once with PBS before adding 500 µl of PBS 1X/4% PFA for 20 min. Finally, the wells were washed with PBS, and the coverslips were immunolabelled.

### Phage precipitation

Bacterial strains grown overnight on BHI agar plates were adjusted to $OD_{600} = 0.05$ in transduction liquid medium without DNase I and incubated at 37 °C for 2 h. Approximately $2 \times 10^8$ bacteria were transferred to a 2 ml tube. One hundred µl of $10^7$ MDA$_{(orf6-aadA1)}$ phage/µl solution or MDA$_{(orf6::aph3)}$ phage/µl solution was added to the bacteria, and the tubes were filled up to 1.4 ml with transduction liquid medium (without DNase I). The mix was incubated for 30 min at 37 °C, then centrifuged at $400 \times g$ for 4 min, 800 µl of supernatant was removed and centrifuged again for 3 min (to pellet bacteria that have settled to the sides of the tubes). The supernatant was removed and the pellet was resuspended in 800 µl PBS 1X in a 1.5 ml tube. The suspension was centrifuged again at $400 \times g$ for 4 min, 600 µl of supernatant was removed, centrifuged again for 3 min, and finally the pellet was resuspended in 100 µl of PBS 1X/4% PFA (paraformaldehyde; Life Technologies). Twenty µl of bacteria in PBS/4% PFA were spread on a slide and dried for 2 h before immunolabelling.

### Phage/pili labelling and quantification

The slides or coverslips were washed with PBS/0.1 M NH$_4$Cl for 2 min and incubated with PBS/4% BSA for 5 min. MDA$_{(orf6-aadA1)}$Φ and MDA$_{(orf6::aph3)}$Φ were stained with the rabbit anti-ORF4 antibody used at 1/100 dilution. For tests with strains expressing PilE$_{SB}$ and PilE$_{SE}$, pili were stained with mouse anti-PilE antibody 20D9, used at 1/1000 dilution. Secondary antibodies were goat anti-rabbit antibody conjugated with Alexa Fluor 488 (Molecular Probes) used at 1/400 dilution, and goat anti-mouse antibody conjugated with Alexa Fluor 546. Bacteria were detected by DNA staining using 100 ng/ml DAPI (49,6-Diamidine-29-phenylindole dihydrochloride). Coverslips were mounted in Mowiol. Images were captured using a Zeiss Apotome fluorescence microscope and images were collected and processed using the ZEN (ZEISS Efficient Navigation) software. MDA binding to bacteria and pili quantity were estimated by immunofluorescence as the ratio of phage surface or pili surface labelling to bacterial surface labelling estimated by staining bacterial DNA with DAPI (arb. units: arbitrary unit).

### Stimulated-emission-depletion (STED) microscopy

Bacteria with or without bound phages in PBS/4% PFA were spread on 22-mm high-precision cover glasses with thickness of 170 µm, (No. 1.5H, Marienfield) and dried for 2 h. The slides were then washed with PBS/0.1 M NH$_4$Cl for 2 min and blocked with 4% BSA–PBS for 10 min, labelled with monoclonal anti-PilE antibody 20D9 (1:1000 in 4% BSA–PBS) and monoclonal anti-ORF4 antibody (1:100 in 4% BSA–PBS) and then incubated with goat anti-mouse Alexa Fluor 514 secondary antibody fragments (1:1000) and goat anti-rabbit ATTO550 secondary antibody fragments (1:500). Bacteria were detected by DNA staining

with 100 ng/ml DAPI. Slides were mounted with ProLong Gold antifade reagent (Invitrogen), refractive index: 1.47.

Images were captured on a Leica TCS SP8 STED microscope, controlled by LASX software, equipped with a 660 nm depletion laser and a 100x NA1.4 objective. Images were acquired using a gated hybrid detector.

The images were then deconvolved using the Huygens software with the 'Deconvolution Express, standard profile' parameter.

### Electron microscopy: negative staining and immunogold-labelling

Bacteria in PBS/4% PFA with and without co-precipitated MDAΦ were adsorbed onto Carbon/Formvar-coated 200-square-mesh copper grids (Euromedex, Souffelweyersheim, France) for 15 min. The grids were then washed twice for 1 min with PBS and placed sequentially on drops of the following reagents at room temperature: PBS/0.1 M glycine (10 min), PBS/1% BSA (bovine serum albumin)/5% normal goat serum (15 min), and then on a drop of anti-ORF4 N-terminal antibody diluted 1/50 in PBS/1% BSA (60 min). After three washes in PBS, the grids were placed on a drop of Fab2' goat anti-rabbit conjugated to 10 nm diameter gold (Aurion, Wageningen, The Netherlands) diluted 1/40 in PBS/1% BSA for 60 min. The grids were then washed three times in PBS (1 min), fixed in PBS/1% glutaraldehyde (5 min), washed twice in PBS and twice in distilled water. The grids were then treated with 1% uranyl acetate, air-dried and viewed. Images were acquired using a Jeol 1400 TEM (Jeol, Croissy-sur-Seine, France) operating at 120 keV and equipped with a RIO CMOS camera (Ametek SAS, Elancourt, France).

### Quantification of the proportion of *pilS3* D-region by qPCR

We estimated the proportion of bacteria with the *pilS3* D-region sequence at the *pilE* locus by qPCR. One pair of primers was designed to anneal to the *pilE* locus. One primer was designed in the *pilS3* D-region and one primer on the conserved upstream part of the *pilE* gene (pilS3-F/R). The second pair of primers (pilT-F/R) was designed to anneal to the *pilT* locus, allowing the quantification of the total number of chromosomes. qPCR was performed on genomic DNA using Luna Universal qPCR Master Mix (NEB). All samples were performed in three independent experiments. The cycling conditions were as follows: 95 °C for 1 min, and 40 cycles of 95 °C for 15 s and 60 °C for 40 s on QuantStudio1 (Applied Biosystems). For the standard curve, genomic DNA from ZΔMDA mutS$^+$ ΔrecA pilE$_{SD}$ (considered to contain 100% of bacteria expressing *pilE* D-region S3) was used. Serial dilutions of this DNA were amplified and a standard curve equation was recorded for each primer pair and used to quantify the samples. For each quantification, DNA from pilE$_{SD}$ clonal and pilE$_{Z4}$ clonal strains, consisting of 100% and 0% of bacteria expressing *pilE* D-region S3, respectively, were used as positive and negative controls.

### Bioinformatics

Pilus models were generated by running the amino acid sequences of PilE, PilC1, PilC2 in Alphafold 3 (https://golgi.sandbox.google.com) with general parameters. Several generated PilE models were aligned to subunits of the biological assembly 5KUA using ChimeraX version: 1.7.1. The structures were coloured using the coulombic electrostatic potential command of ChimeraX with the default colouring range. Folding structures of PilC1 and PilC2 were also obtained by running the amino acid sequences in Alphafold 3 with general parameters.

Sequence alignments were performed using Clustal-omega (1.2.4) (https://www.ebi.ac.uk/jdispatcher/msa/clustalo). Isoelectric points of PilE sequences were determined from their PilE amino acid sequences using the tool iep45/5.0.0.1 in Galaxy France.

### Statistical analysis

Statistical analyses were performed using GraphPad Prism 8.02. Taking into account the number of replicates, the normality of the

distribution for the whole data sets was assessed using QQ plots. When necessary, data were log transformed. Homogeneity of variance was tested using the Brown–Forsythe F-test. If the distribution of the data conformed to the law of normality and the variance of the groups was homogeneous, multiple comparison analyses were assessed using a one-way ANOVA with multiple comparisons test and appropriate correction, and data were expressed as mean ± SEM or median ±95% confidence interval (relevant $p$-values were reported in the figures). In the case of different variances, a Brown–Forsythe and Welch ANOVA test was performed. A two-way ANOVA with Sidak correction was performed in the Supplementary Fig. 7. The H0 hypothesis was rejected at a significance level of $p \leq 0.05$.

### Reporting summary

Further information on research design is available in the Nature Portfolio Reporting Summary linked to this article.

### Data availability

Raw data and statistical analysis are listed in the Source Data file. Source data are provided with this paper.

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

## Acknowledgements

This work was supported by the research grants ANR-23-CE15-0004-01 from ANR and ANR-18-IDEX-0001-EMERGENCE from Université Paris-Cité (to EB); INSERM (to MC); Université Paris-Cité (PhD scholarship to CM and MW) and Année de recherche from AP-HP and ARS (île de France) to AB. We warmly thank Dr Laetitia Houot and Dr Callypso Pellegri for tools and advice for the two-hybrid assay. We thank Pr Isabelle Martin-Verstraete for fruitful discussions on the two-hybrid assay. We thank Pr Nicolas Biais for his comments and help all along this work.

## Author contributions

C.M., A.B., H.L., M.C., E.B. designed research; C.M., A.B., M.W., M.M., J.M., C.I. performed research; M.M., J.M., B.D., C.I. contributed reagents/analytic tools; C.M., A.B., M.W., M.M., H.L., A.J., C.I., M.C., E.B. analyzed data; M.C. and E.B. validated data; M.C. and E.B. contributed to funding acquisition; M.C. and E.B. coordinated the research; C.M., M.C., E.B. wrote the paper; C.M., H.L., A.J., X.N., M.C., E.B. reviewed and edited the paper.

## Competing interests

The authors declare no competing interests.
