## [Transparent Peer Review file · Nature Communications]

***Neisseria meningitidis* filamentous phage MDA promotes colonisation by selecting hyperadhesive pili variants**

Corresponding Author: Dr Emmanuelle Bille

Version 0:

Reviewer comments:

Reviewer #1

(Remarks to the Author)

The manuscript by Mouville et al is a creatively devised study designed to investigate the role of a filamentous bacteriophage in meningococcal biology. It is a clear extension of the research groups previous work on the meningococcal MDA phage biology. In some of this groups previous work, they have shown that the MDA bacteriophage is able to infect different *N. meningitidis* strains, using the type IV pili as a receptor, and that infection requires PilE and PilT (<https://doi.org/10.1099/mic.0.000215>). In this study, they explore how changes in PilE sequence impact the ability of the phage to infect the meningococcus and propose mechanisms by which the changes in infectivity could impact bacterial virulence. Their data clearly demonstrate how antigenic variants of PilE can be selected for using the MDA phage and that these variants are better able to adhere to epithelial and endothelial cells.

General comments

While the data is presented in a logical fashion, because of how the transduction data is presented (normalized to Z5463ΔMDA), it is difficult to interpret the some of the data. Figure 1a-d clearly demonstrates how various pilin-associated proteins impact transduction frequency. However, Imhaus and Duménil (<https://doi.org/10.15252/emj.201488031>) previously showed that a meningococcus has about 6 pili per bacterium, and a pilX mutant has a three-fold reduction in pili. This is similar to the change in quantity that the authors report in figure 1c. On line 168, the authors state "The ΔpilC1/C2 double mutant is barely piliated". However, their data indicates that there is only about a 3 fold reduction. This is not barely piliated (see also line 227).

In figure 1e, the authors do not define the PilE sequence of ZΔMDA (line 179). This should be in supplemental figure 1). Supplemental figure 2 does not have this sequence, or pilEsb, and this also needs to be added here. In figure 1, they do not define the limit of detection for a transduction experiment? If PilE is truly required for transduction, it would seem that the transductants identified in figure 1 A for PilE represent the mutation frequency to spectinomycin resistance. Is this true? Otherwise, it suggests that PilE is not required for transduction but enhances transduction. If one assumes that PilE expression is needed for transduction, and the starting concentration of bacteria in their experiments is 10⁹ CFU/ml (about an OD of 0.6), then the efficiency of transduction in this figure is 10⁻⁵. This becomes important when trying to interpret the data in Figure 6 (see below).

In figure 2, panel b, when the authors sequenced 10 random colonies, 50% underwent phase variation. This is much higher than that that has been reported for the meningococcus (doi: 10.1128/JB.00465-18). How do the authors explain this extraordinarily high rate?

In figure 3, it would be useful to know how the truncated PilC proteins are impacting surface charge. If there is no change, this would strengthen their argument in the data they present in figure 5.

In figure 5, the authors have assumed that small changes in pilin, which change the PI of pilin, impact the surface charge of the bacteria. It is possible that other phase variable surface molecules are also changing (Opa as an example)? The authors need to demonstrate that the amino acid changes are contributing to surface charge changes, perhaps by measuring the Zeta potential of the various strains.

In figure 6, panel B it would appear that pilEs3 represents about 0.1% of the starting inoculum. After incubating with cells there is about a 10 fold enrichment for selecting the spontaneous phase variant. However, in figure 3 above the change is 50%. How do the authors explain this discrepancy? In panel D, given the low percentage of pilEs3, and the low transduction efficiency, it is not surprising that virtually all of the transduced bacteria are pilEs3 positive (about 4 logs). What is surprising is that there is only a 1 log increase in adherence. Doesn't this argue that the presence of the phage interferes with adherence?

Bille et al have previously show that phage expression promotes bacterial aggregation

(<https://doi.org/10.1371/journal.ppat.1006495>). In figure 7, if the presence of phage promotes bacterial aggregation, such, adhering bacteria containing phage would be represented by microcolonies that would be larger than those lacking the phage. One might explain the modest increase in the presence of phage in adhering bacteria, as seen in figure 7 panel B as simply the result of this. Do the authors have confocal data to show that the control and adherent data are producing the same size microcolonies on cells?.

In the discussion, the authors try to link pilin expression with virulence. It is hard for this reviewer to understand the logic. This relationship may exist the first time a bacterium becomes infected with MDA, but subsequent replication in a host would allow for antigenic variation where the strain no longer expresses the PilE needed for phage infection. Unless the strain is constantly being reinfected with MDA, how would phage infection be linked to virulence. Do the authors have any indication of this?

Specific comments

Line 162. that the phage only fits into strains expressing fully retractable pili. I think the authors mean “that the phage only infects strains expressing fully retractable pili.”

Line 215. Overall, these results suggest that MDA Φ : (i) infects meningococci expressing a specific PilE variant. This really isn't correct. The data really says (in agreement with their overall interpretation) that some variants appear to be more susceptible to infection.

In discussing the data in figure 2, the authors have missed an opportunity to use their data to conclude that phage infection is recA independent

Line 410. The DNA sequence of strain Z5463 is known: hence its pilS variability is also known.

On line 411-12, they indicate. “This allows us to observe the preferential integration of phages into the genome of *N. meningitidis* expressing high pI PilE variants, although this variant is expressed by a minority of bacteria. It is unclear what the highlighted phrase means.

In figure 4b, the data for each replicate needs to be visualized.

In figure 5, it would be helpful if the legend stated that Δ MDA was expressing pilESD

In figure 6, panels B and D. Were the PCRs performed on the three independent experiments or 3 times on the same experiment.

In supplemental figure 2 it would be helpful to have the DNA sequence of the PilE z1-4 strains.

In supplemental figure 4, the colors of the first three columns are white. Legend needs to explain this.

In the table of strains, the construction of the Δ MDA strain has¹ the wrong reference. It should be *J Exp Med* (2005) 201 (12): 1905–1913.

Reviewer #2

(Remarks to the Author)

A filamentous phage promotes meningococcal colonization by selecting hyperadhesive pili variants linking phage infection to bacterial virulence

This is a manuscript showing that MDA phage aids in selection of *N. meningitidis* variants with superior adhesion to the host cells and thereby is bound to increase virulence during the infection of a host. This is an interesting but not novel concept when it comes to filamentous phages and their role in virulence of their hosts.

The manuscript is generally very well written, however important filamentous bacteriophage biology published findings that are relevant to this manuscript has been missed, both from the recent years and from the past.

There are a few methodological gaps in the manuscript and some experimental findings that require clarification, proper controls or statistical analyses.

Major comments

L233-234 “PilC Δ 3-843 allowed transduction and transformation at a level equivalent to 10% of the control (Fig. 3b,c).”

Isn't this an indication that the C domain of PilC may be involved in binding of MDA?

L242 Supplementary Fig. 3b:

This figure describes bacterial adenylate cyclase two-component assay which is here used to rule out the interactions of Orf6 with minor pilins. Unfortunately there is no appropriate positive control showing that the Orf6 fusions used in this experiment have folded into a functional form. This is crucial because the type of two-hybrid system used in this experiment means that the proteins have to fold correctly in the cytoplasm. Given that the tested proteins fold in the periplasm prior to assembly into the phage/pili, and have S-S bridges that are required for their correct folding (which do not form in the cytoplasm), it is quite possible that none of the pilins or Orf6 in this experiment are folded into functional domains. In the absence of Orf6 and pilin positive control interactions, negative results shown in this figure cannot discern between failed interactions and misfolding of one or both partners.

The only positive control is pIII from M13 phage (M13 pIII - ToIA interaction gives the only positive signal in the figure). Why was MDA Orf6 interaction with -N.m. ToIA not tested in this setup? Why were no interactions with pilE tested, given the manuscript is focused on showing that PilE is the key receptor for MDA?

L250-251 + Fig. 4. “we first performed transmission electron microscopy (TEM) on whole bacteria incubated with phages “
The TEM shows MDA longitudinally associated with the pilus, through pVIII-pilE interactions. This may be a secondary step after the initial binding of Orf6 to either pilE or minor pilins, or both, and may strongly contribute to the overall binding. Competition studies using purified MDA Orf6 or MDA phage that has truncated pIII (without the host-binding domains) are required to resolve the pairing i.e. does pIII first interact with -pilC(C-terminal domain), then pVIII interacts with pilE? or Orf6-pilE concurrently with pVIII-pilE?

L251-257/ Fig. 4a; Supplementary Fig. 4: Why is the negative control shown in supplementary figure? Side by side comparison would be more impactful. Was there any statistical analysis of gold bead pilus-association done on multiple

cells? This would strengthen the validity of observations.

Fig. 6-7: Q - why are the values shown in 6c and 6f-g relative (% relative to the value of the control sample), while absolute value is shown in 7b?

L461 "10% NaCl and 20% polyethylene glycol 6000" – Is this the stock concentration or the final concentration? This is way too high as a final concentration for precipitation (normally it is 0.5 M NaCl, 5% PEG for filamentous phages). If only one round of PEG precipitation was done and no further purification steps were performed, the samples would be highly contaminated with host proteins and cellular debris or vesicles such as OMVs.

462-463 "resuspended in PBS 1X (with CaCl₂ and MgCl₂; Gibco) and used directly"

In the presence of divalent cations OM fragments or vesicles will not be soluble. Was the insoluble pellet removed from the phage stock by centrifugation, to eliminate OMVs?

L467-468 "Two hundred microliters of the bacterial suspension were aliquoted into a 24-well plate and mixed with 30 µl of the MDA phage preparation."

Were the phage stocks titrated? How many phages were used per transduction? What was the m.o.i.? Were there any attempts to quantify phages by densitometry or spectrophotometry? This is also essential to ensure that the same amount of phages is always used between different experiments. See Waldor and Mekalanos 1996

<https://www.science.org/doi/10.1126/science.272.5270.1910> (Table 1) and publications thereafter.

Same below:

L475-479 "The efficacy of the phage preparation was verified by transducing 30 µl of the phage preparation into the ΔMDA strain and validated if the transduction frequency was greater than 10⁻³. If no bacteria were obtained at the least diluted point on the agar plates, a CFU of 0.9 was arbitrarily assigned and the point was marked as below the detection limit.

Same comment as above: how many phages were used, i.e. what was the m.o.i.?

L607-609 Quantification of pili by western blotting:

What was used to plot a standard curve here? Standard should be serial dilutions of a purified pilin subunit quantified by a colorimetric or other absolute protein quantification method. Serial dilutions of quantified pilin subunits should be run on the same SDS-PAGE gel along with the pili samples, transferred and used to construct a standard curve so that the extracted pilin bands in the westerns can be correctly quantified.

L612 "Phage precipitation" method

This looks like a protocol for detecting bacteria-associated vs. free phages. The assay commences by mixing phage and bacteria at a huge excess of phages (10¹¹ phages per 2x10⁸; i.e. m.o.i. 500). Since the efficiency of transduction is well below 100% based on data in the manuscript (albeit absolute numbers are mostly missing), this assay where there is huge excess of phages over the host bacteria is puzzling.

It is good to see that the phages were quantified before use in this experiment. Method of quantification, however, was not described. Titer vs. quantity of the phages seem to be strongly dependent on the strain, hence it is really important to measure the number of phage particles (quantity of phage) in addition to titration based on host infection.

Given the very crude MDA phage purification method is described earlier in the methods section, which results in phage preparations containing large amount of contaminating proteins derived from the host (L451-463; also see a comment above), it would not be advisable to use simple spectroscopy to quantify phages. Instead, quantification of phage DNA would be a much more accurate measure of the number of phages per volume of the sample. Methodology for phage DNA quantification is well-established in the phage literature (densitometry or qPCR).

Minor comments

L35-36 F pilus is not a type IV pilus. Rewrite the sentence:

The paradigm of filamentous phage infection has been defined for the phages Ff and CTX, where binding to the tips of bacterial type IV pili is a means of infecting their bacterial host

L42-44 Taken together, this study reveals how the propagation strategy of a filamentous phage, aimed at selecting the best coloniser as a host, can promote the selection of hyperadhesive bacterial variants, linking phage infection to bacterial virulence.

Convoluting sentence. Simplify.

L62-63 The first filamentous phage of this family was described by Peter H. Hofschneider in *Escherichia coli* in the 1960s and named M13 (Ray et al., 1966).

This is not the first report, look for earlier references for Ff

Hofschneider PH. 1963. Untersuchungen über kleine *E. coli* K 12 Bakteriophagen 1 und 2 Mitteilung. *Z Naturforsch Pt B* 18: 203–210. doi:10.1515/znb-1963-0306

Loeb T. 1960. Isolation of a bacteriophage specific for the F⁺ and Hfr mating types of *Escherichia coli* K-12. *Science* 131: 932–933. doi:10.1126/science.131.3404.932

Marvin DA, Hoffmann-Berling H. 1963. A fibrous DNA phage (Fd) and a spherical RNA phage (Fr) specific for male strains of *E. coli* ii. Physical characteristics. *Z Naturforsch B* 18: 884–893. doi:10.1515/znb-1963-1106

L63 replace "famous" with "well-known"

L64 replace "CTX" with "CTXφ"

L66 "Their main features are the mutually beneficial trait conferred to their host"

Ff are not beneficial to the host. They slow the cell cycle and cause envelope stress. Rewrite.

L68-75 Paragraph reads as if only Pf4, CTX and MDA filamentous phages have been reported to have an effect of bacterial host. However, there are a number of filamentous phages that infect *Pseudomonas*, *Vibrio* and plant pathogenic bacteria such as *Xanthomonas* etc. that have been reported to have impact on bacterial virulence.

L76-L77 "Almost all filamentous phages described so far infect diderm bacteria and must therefore cross both the outer and inner membranes to infect their target."

This is no longer correct. There are over 10,000 filamentous phage-derived prophages in the current sequence databases. Many gram-positive bacteria host filamentous phages, as well as some archaea. See Roux S, et al. 2019. Cryptic inoviruses revealed as pervasive in bacteria and archaea across Earth's biomes. *Nat Microbiol* 4: 1895–1906. doi:10.1038/s41564-

019-0510-x (and the follow-up articles).

Rewrite: Filamentous phages that infect diderm bacteria must cross both the outer and inner membranes to infect their target. L80 “such as Ff and related phages that use the sexual/conjugative pilus F of *E. coli* (Hay and Lithgow, 2019).”

Cited paper is a general review on Ff phages. There are much more appropriate primary research references for this. Use more appropriate references.

L86-87 “pilus porin”

Type IV pilus assembly system does not have a porin. Its outer membrane component pilQ is a channel of the secretin family that is unrelated to porins.

Replace with the correct term.

L118-119, comment: same as above. PilQ is not porin, it is a channel of the secretin family.

L138 – comment: binding of filamentous phages to the tip of the pilus is not a paradigm. For example, filamentous phages infect unpiliated bacteria (e.g. *Yersinia pestis* filamentous phage CUS-1–

<https://www.sciencedirect.com/science/article/pii/S0042682210005064>)

L160-161 “Transduction was impaired in mutants with low pili quantity and low transformation efficiency, with the exception of the Δ pilT mutant, confirming ...”

Unclear sentence. It could be read as if transduction is not impaired in Δ pilT mutants. Rewrite.

L165 “these proteins are poorly involved in phage transduction.”

Rewrite: “these proteins have a minor role in phage transduction”

Additional comment: Transduction is used as an assay for MDA infection, so can be referred to as infection.

L203-204 “To confirm that antigenic variation in the Δ MDA pilESA strain is sufficient to explain transduction by MDA ϕ , we constructed...”

L220-222 “The near absence of phage infection in the Δ MDA mutS+ Δ recA pilESA strain seems to disqualify the pili tip as an MDA ϕ receptor, although it is considered to be the receptor for filamentous phages f1 and CTX (Jacobson, 1972; Gutierrez-Rodarte et al., 2019).”

Clumsy sentence (“disqualify”); repetition the statement about the pilus tip is unnecessary. Rewrite.

229-230 “the first 555 amino acids of PilC1”

It cannot be “first”; it can only be “N-terminal” 555 amino acids. However according to the figure, deletion is from residue 3 to 555, inclusive. Rewrite.

230 Spell out N-terminal domain, or define “Nter”. In the figures there is also a term “N-ter”

231-232 “Deletion of more than the first 555 amino acids reduced ...”

Change to

Increasing deletion to 3-843 residues reduced ...

Throughout: Subscripts and superscripts were not included in chemical formulae: MgCl₂, MgSO₄ – check and correct throughout.

Reviewer #3

(Remarks to the Author)

Summary: In this study, the authors build upon their past work into the role of MDA ϕ in *N. meningitidis* colonization and biology. The authors confirm and expand their earlier data that Tfp biogenesis is critical for MDA phage transduction and that MDA ϕ uses its capsid protein to adhere to PilE along the length of the Tfp. The authors find that MDA ϕ preferentially binds to and infects bacteria expressing positively charged PilE variants and that phage transduction not only enhances host colonization but host colonization also enhances phage transduction. This work adds important knowledge to the critically understudied filamentous phage literature and represents one of the first (to our knowledge) connections between Tfp antigenic variation and phage infection. However, the authors need to explain better the effect of the deletion of each Tfp component on pilus assembly and morphology.

Major points:

1. All of the PilE variants (Figure 5) have higher transduction efficiencies than pilESA. How representative is the reduced charge of pilESA in primary isolates?
2. The authors should list or discuss the T4p morphology of all the mutants; PilE quantity alone does not indicate piliation. Doing this will clarify the connection between pilus biogenesis and phage transduction. For instance, some of the pilin deletions (i.e., pilW) appear to have an intermediate transduction phenotype, which could be due to altered piliation.
3. Others have shown that tfpC deletion in *N. meningitidis* and *N. gonorrhoeae* affects Tfp biogenesis, pilin quantity, and transformation efficiency (Muir et al. *Nature Communications*, 2020; Hu et al. *mBio*, 2020). In this study, a Δ tfpC mutant displayed no difference in transformation efficiency and pilin quantity. The authors should discuss this discrepancy.
4. While they show that PilE is critical for phage transduction, the authors should be careful not to deemphasize the potential role of the other Tfp components in this process. For example, PilQ may also be important for MDA phage transduction, but its loss also results in non-piliated bacteria. Thus, the loss of piliation could be confounding the effect of pilQ deletion in phage transduction.

Minor Points:

1. Line 58: Caudovirales should be italicized.
2. Line 66: Please clarify the “mutually beneficial trait”.
3. Line 69: Please provide examples of the environmental signals that trigger phage gene induction.
4. Line 140: Please choose a different word from “benefits” or clarify that the phage benefits from the positive charge.
5. Figure 1: List or discuss the piliation states of each mutant.

6. Figure 1: Reconfigure the graphs so that the transduction, transformation, and pilin quantities are all next to each other for each gene deletion. The colors of each bar could be different for each measurement.
7. Figure 1 A-C needs statistics.
8. Lines 205-206: this statement needs a citation.
9. Line 215: Re-write this for "better infects certain variant", as the phage can still infect other pilE variants, just at lower efficiency
10. Line 228: Why was PilC1 chosen instead of pilC2?
11. Please speculate why a pilE deletion retains some transduction and transformation efficiency.
12. Line 265: it looks like some phage did bind to PilESA, just at reduced levels
13. Supplemental Figure 4e: What is the MDA binding to in the Δ pilE strain? It is unclear how they calculated binding. This should be clarified in lines 280-282
14. Supplemental Figure 3: all these experiments test Orf6 binding to the pilus fiber proteins. Why was Orf4 not tested, as this protein was shown to be responsible for PilE binding?
15. Figure 5b: clarify that the reference strain (Δ MDA) carries the PilESD variant. It is unclear from this graph.
16. Line 318-319: Why 70-71 in D region was analyzed is unclear. Are these charges not on the surface? Please justify.
17. Figure 6B: The % positive and negative bacteria with the pilES3 variant should be separate bars. This graph is deceiving because it appears that % pilES3 positive bacteria are at 50%, when it is only 1%.
18. Figure 6: Clarify that for figures A-C, transduction occurs after colonization, while for figures D-F, transduction occurs before colonization
19. Figure 6c should have the same Y-axis scale as Figures 6f-g.
20. Can you comment on how widespread the MDA phage is in N. meningitidis?
21. Figure 7C: This image suggests that the MDA phage improves transmission, but the data do not necessarily support this.
22. Can MDA phage bind the Class II PilE and infect?
23. Is it possible that only a few pili are necessary? Some of these mutants may appear non-piliated but may have a few pili capable of facilitating transduction.

Reviewer #4

(Remarks to the Author)

Version 1:

Reviewer comments:

Reviewer #1

(Remarks to the Author)

In figure 1e, the authors do not define the PilE sequence of Δ MDA (line 179). This should be in supplemental figure 1.

Supplemental figure 2 does not have this sequence, or pilEsb, and this also needs to be added here.

We agree with Reviewer #1. However, the pilE gene of Δ MDA cannot be determined, since the strain corresponds to a mix of pilE sequences, with no dominant one having been determined.

I do not understand this response. The strain has been manipulated in the laboratory to lack phage. This would indicate a selection event. Therefore, even if the parent from the original case was diverse, the mutant should be clonal. This information is needed to understand the relationship between the parent and the mutant pilis that they are analyzing. What if the PilD region of the parent is the same as PilESA or PilE SB?

We have added this information to line 127.

In figure 1, they do not define the limit of detection for a transduction experiment? If PilE is truly required for transduction, it would seem that the transductants identified in figure 1a for PilE represent the mutation frequency to spectinomycin resistance. Is this true?

We thank the reviewer for this comment. Indeed, it is known that filamentous phages interact with pili, allowing them to be pulled in by the pili secretin (Hay et al., EMBO reports 2019 <https://doi.org/10.15252/embr.201847427>).

This reference does not show that the phage interact with the gonococcal pilin secretin

Here, we observed that transduction in the Δ pilE mutant is 10,000 times lower than in the wild type. The same is true for the Δ pilQ mutant. Therefore, we can assume that any pili-defective mutant will have a transduction frequency below the detection threshold, and that colonies growing in these mutants will be spectinomycin-resistant colonies due to spontaneous mutation (note that spectinomycin resistance is due to a mutation in the 16S rRNA: <https://doi.org/10.1128/aac.44.5.1365-1366.200>). We have therefore added the background threshold to the figure 1.

The establishment of background is based on a series of assumptions. If they are transductants, they should have

the spec cassette. They should use this to determine the detection level.

In figure 2, panel b, when the authors sequenced 10 random colonies, 50% underwent phase variation. This is much higher than that that has been reported for the meningococcus (doi: 10.1128/JB.00465-18). How do the authors explain this extraordinarily high rate?

We agree with Reviewer #1 and were also surprised. However, this rate must be considered in relation to the number of generations that have passed since the PilesA variant was introduced to the original strain (at least two overnight cultures on plates). We observed the same variant rate with a PilesB variant.

This suggests that the starting cultures had mixed genotypes. Furthermore, the strain Z5463 has a mutS mutation (Omer et al., PLoSOne 2011, doi.org/10.1371/journal.pone.0017145), which may explain the high rate of punctual mutations observed in three strains of the ten strains. **If the authors explanation is correct, why did they see the same pilin mutations and not a variety?.**

In the discussion, the authors try to link phage expression with virulence. It is hard for this reviewer to understand the logic. This relationship may exist the first time a bacterium becomes infected with MDA, but subsequent replication in a host would allow for antigenic variation where the strain no longer expresses the Pile needed for phage infection. Unless the strain is constantly being reinfected with MDA, how would phage infection be linked to virulence. Do the authors have any indication of this?

The reviewer #1 raises an important question. The presence of an average of two to four prophages per N. meningitidis strain indicates that prophages result from reinfection rather than duplication (Kawai et al., DNA Research 2005, doi: 10.1093/dnares/dsi021, Al Suwayyid et al., GBE 2020, doi:10.1093/gbe/evaa023) and thus bacteria should be constantly reinfected.

We now discuss this point in the text (lines 409-411).

These references do not really address the question. One of the references suggests that phage acquisition is by transformation.

Reviewer #2

(Remarks to the Author)

Authors have taken some steps to improve the quality of writing and data, however manuscript requires more work on both fronts.

The most critical issue is that number of MDaphi used in binding and transduction experiments is not given. Rather it was generally stated that purified phage numbers were "10E7 per microliter at average". This gives quite a low phage:bacteria ratio. Given that most binding assay results are given in relative values, the veracity of findings is hard to gauge.

Reviewer #3

(Remarks to the Author)

The authors have responded well to the previous three reviews.

Version 2:

Reviewer comments:

Reviewer #1

(Remarks to the Author)

The authors have addressed all of my concerns

RESPONSES TO REFEREES

We would like to thank the reviewers for their positive feedback on our work and for their comments. As requested by the editor, we have addressed the reviewers' concerns and modified the manuscript and figures in the revised version accordingly. These changes are presented point-by-point below.

Following editorial requests, we have reduced the size of the abstract and proposed a shorter title: "The meningococcal disease-associated phage promotes meningococcal colonisation by selecting hyperadhesive pili variants".

Line numbers correspond to the document « Mouville_et_al_manuscript_Marked-Up »

Reviewer #1 (Remarks to the Author):

While the data is presented in a logical fashion, because of how the transduction data is presented (normalized to Z5463 Δ MDA), it is difficult to interpret some of the data. Figure 1a-d clearly demonstrates how various pilin-associated proteins impact transduction frequency. However, Imhaus and Duménil (<https://doi.org/10.15252/emj.201488031>) previously showed that a meningococcus has about 6 pili per bacterium, and a *pilX* mutant has a three-fold reduction in pili. This is similar to the change in quantity that the authors report in figure 1c. On line 168, the authors state "The Δ *pilC1/C2* double mutant is barely piliated". However, their data indicates that there is only about a 3-fold reduction. This is not barely piliated (see also line 227).

We understand that our method of quantifying pili was misleading for the three reviewers. It is based on a method of pili extraction and purification that was previously published in PNAS in 2021 (<https://doi.org/10.1073/pnas.2109364118>). We agree that the sensitivity of this assay is very low. Consequently, the background signal appeared to be at levels up to 5% of those observed in the wild type. As suggested by reviewer #1, we have proposed a new strategy for pili quantification based on imaging of assembled pili: Imhaus and Duménil (<https://doi.org/10.15252/emj.201488031>). Unlike the method proposed by Imhaus et al., we did not quantify the number of pili, which requires the observation of isolated bacteria, but rather the total surface area of pili detected by immunofluorescence, normalised to the surface area of bacterial DNA detected by DAPI. Using this strategy, we demonstrate that the *pilC1/pilC2* double mutant exhibits piliation similar to that observed in *pilE* or *pilQ* deficient mutants – which was expected. Figure 1c and its legend have been modified accordingly (lines 739-755). The material and method used to construct the strains used for immunofluorescence and the quantification itself have been added (lines 582-585 and 642-646).

In figure 1e, the authors do not define the PilE sequence of Z Δ MDA (line 179). This should be in supplemental figure 1.

Supplemental figure 2 does not have this sequence, or *pilEsb*, and this also needs to be added here.

We agree with Reviewer #1. However, the *pilE* gene of Z Δ MDA cannot be determined, since the strain corresponds to a mix of *pilE* sequences, with no dominant one having been determined. We have added this information to line 127.

We have also added the nucleic and protein sequences of *pilE_{SB}* to supp Fig 2a and 2b.

In figure 1, they do not define the limit of detection for a transduction experiment? If PilE is truly required for transduction, it would seem that the transductants identified in figure 1a for PilE represent the mutation frequency to spectinomycin resistance. Is this true?

We thank the reviewer for this comment. Indeed, it is known that filamentous phages interact with pili, allowing them to be pulled in by the pili secretin (Hay *et al.*, EMBO reports 2019 <https://doi.org/10.15252/embr.201847427>). Here, we observed that transduction in the Δ *pilE* mutant is 10,000 times lower than in the wild type. The same is true for the Δ *pilQ* mutant.

Therefore, we can assume that any pili-defective mutant will have a transduction frequency below the detection threshold, and that colonies growing in these mutants will be spectinomycin-resistant colonies due to spontaneous mutation (note that spectinomycin resistance is due to a mutation in the 16S rRNA: <https://doi.org/10.1128/aac.44.5.1365-1366.200>). We have therefore added the background threshold to the figure 1.

Otherwise, it suggests that PilE is not required for transduction but enhances transduction. If one assumes that PilE expression is needed for transduction, and the starting concentration of bacteria in their experiments is 10^9 CFU/ml (about an OD of 0.6), then the efficiency?? of transduction in this figure is 10^{-5} . This becomes important when trying to interpret the data in Figure 6 (see below).

Indeed, we did not mention the quantity of phage per μl added for transduction experiment in the original manuscript. The transduction frequency was mentioned in line 483: "The efficacy of the phage preparation was verified by transducing 30 μl of the phage preparation into the ΔMDA strain and was validated if the transduction frequency was greater than 10^{-3} ", but in the absence of phage quantity this sentence was misleading. We have now added the number of phages per μl of phage preparation (assessed by qPCR) (lines 461-470). Considering this value ($2 \cdot 10^7$), we can assume that the transduction efficiency is 10^{-3} when 30 μl of phages are mixed with 200 μl of bacteria resuspended at an OD_{600} of 1 in transduction liquid medium (M.O.I. = 3).

In figure 2, panel b, when the authors sequenced 10 random colonies, 50% underwent phase variation. This is much higher than that that has been reported for the meningococcus (doi: 10.1128/JB.00465-18). How do the authors explain this extraordinarily high rate?

We agree with Reviewer #1 and were also surprised. However, this rate must be considered in relation to the number of generations that have passed since the PilE_{SA} variant was introduced to the original strain (at least two overnight cultures on plates). We observed the same variant rate with a PilE_{SB} variant. Furthermore, the strain Z5463 has a *mutS* mutation (Omer et al., PLoSOne 2011, doi.org/10.1371/journal.pone.0017145), which may explain the high rate of punctual mutations observed in three strains of the ten strains.

In figure 3, it would be useful to know how the truncated PilC proteins are impacting surface charge. If there is no change, this would strengthen their argument in the data they present in figure 5.

We agree with Reviewer #1. The T4aP model propose that one PilC/PilY1 protein is exposed at the tip of each pilus (Treuner-Lange *et al.* Nature Communications 2020, DOI: 10.1038/s41467-020-18803-z). Therefore, changing the charge of PilC does not affect the overall charge of the pili, which are formed by hundreds of PilE proteins, nor that of bacteria themselves.

In figure 5, the authors have assumed that small changes in pilin, which change the PI of pilin, impact the surface charge of the bacteria. It is possible that other phase variable surface molecules are also changing (Opa as an example)? The authors need to demonstrate that the amino acid changes are contributing to surface charge changes, perhaps by measuring the Zeta potential of the various strains. We respectfully disagree with Reviewer #1. Our interest lay in the surface charge of pili rather than that of whole bacteria. Our hypothesis is that phages first interact with pili, which should occur at a distance up to several micrometers from the bacterial body - as suggested by Figure 4 where phages are not observed in the direct vicinity of bacterial bodies. We could assume that changes in pili PI will not modify the overall PI of the bacterial outer-membrane.

We have discussed this matter with Pr Nicolas Biais, an expert in T4P physics (PMID: 21993648). He confirms that pili are a very long structures whose charge is independent of the bacterial surface charge.

However, since this was not demonstrated we dampened our statement in the discussion line (399-403): “One hypothesis that explains the association of the MDA phages with these PilE variants is electrostatic attraction between the positively charged pilin proteins and the bacteriophages whose capsids are mainly negatively charged, as recently shown by Boehning and collaborators using cryoEM (Boehning et al., 2024), although we cannot exclude an interaction between the phage capsid and specific amino acids on PilE.”

In figure 6, panel B it would appear that pilEs3 represents about 0.1% of the starting inoculum. After incubating with cells there is about a 10-fold enrichment for selecting the spontaneous phase variant. However, in figure 3 above the change is 50%. How do the authors explain this discrepancy?

We think that Figure 2 - we thought that Reviewer #1 referred to as Figure 2 - and Figure 6 are not comparable. In Figure 2 we observed that the PilE_{SA} variant was replaced by other PilE variants, which were not quantified. In Figure 6, we have quantified that 0.1% of the strain expresses *pilES3*, but we did not quantify PilE_{Z4} variant stability at *pilE* locus. In Figure 2, we did not quantify the presence of *pilS3* in the Δ MDA *pilE*_{SA} population. We made this quantification, finding that only 0.03% of the Δ MDA *pilE*_{SA} population expresses *pilS3*. We did not consider it necessary to include this result in the manuscript so as not to make it too lengthy, but if the reviewer and editor feel it is necessary, we will be happy to add it.

In panel D, given the low percentage of pilEs3, and the low transduction efficiency, it is not surprising that virtually all of the transduced bacteria are pilEs3 positive (about 4 logs). What is surprising is that there is only a 1 log increase in adherence. Doesn't this argue that the presence of the phage interferes with adherence?

As mentioned by reviewer #1, Figure 6e is a control experiment and the result was expected. Therefore, most strains expressed *pilES3* after transduction and selection. However, the 1 log increase in adhesion is not unexpected and reflects the difference in adhesion between PilS3 and other PilE variant that should be adherent. Furthermore, we did not observe differences of adhesion between bacteria expressing or not the MDA phage (see below, next comment).

Bille et al have previously show that phage expression promotes bacterial aggregation (<https://doi.org/10.1371/journal.ppat.1006495>). In figure 7, if the presence of phage promotes bacterial aggregation, such, adhering bacteria containing phage would be represented by microcolonies that would be larger than those lacking the phage. One might explain the modest increase in the presence of phage in adhering bacteria, as seen in figure 7 panel B as simply the result of this. Do the authors have confocal data to show that the control and adherent data are producing the same size microcolonies on cells?

The reviewer #1 raises an important question. However, S4 Figure in the paper by Bille et al. 2017 (DOI: 10.1371/journal.ppat.1006495) shows that adhesion at an early time point had been estimated for a strain and an isogenic mutant deleted for MDA, with no differences observed.

In the discussion, the authors try to link pilin expression with virulence. It is hard for this reviewer to understand the logic. This relationship may exist the first time a bacterium becomes infected with MDA, but subsequent replication in a host would allow for antigenic variation where the strain no longer expresses the PilE needed for phage infection. Unless the strain is constantly being reinfected with MDA, how would phage infection be linked to virulence. Do the authors have any indication of this? The reviewer #1 raises an important question. The presence of an average of two to four prophages per *N. meningitidis* strain indicates that prophages result from reinfection rather than duplication (Kawai et al., DNA Research 2005, doi: 10.1093/dnares/dsi021, Al Suwayyid et al., GBE 2020, doi:10.1093/gbe/evaa023) and thus bacteria should be constantly reinfected. We now discuss this point in the text (lines 409-411).

Specific comments

Line 162. that the phage only fits into strains expressing fully retractable pili. I think the authors mean “that the phage only infects strains expressing fully retractable pili.” We have modified the text accordingly (line 133).

Line 215. Overall, these results suggest that MDA Φ : (i) infects meningococci expressing a specific PilE variant. This really isn't correct. The data really says (in agreement with their overall interpretation) that some variants appear to be more susceptible to infection. We have modified the text accordingly (line 187).

In discussing the data in figure 2, the authors have missed an opportunity to use their data to conclude that phage infection is recA independent. We have modified the text accordingly (line 180).

Line 410. The DNA sequence of strain Z5463 is known: hence its *pilS* variability is also known. The sentence has been rewritten (line 396). The diversity of pilE variant of the Z5463 strain is due to antigenic variation and probably the hypermutator phenotype of the strain (Omer *et al.*, 2011, DOI: 10.1371/journal.pone.0017145).

On line 411-12, they indicate. “This allows us to observe the preferential integration of phages into the genome of *N. meningitidis* expressing high pl PilE variants, although this variant is expressed by a minority of bacteria. It is unclear what the highlighted phrase means. The sentence was misleading and has been rewritten (line 397).

In figure 4b, the data for each replicate needs to be visualized. The different replicates were colour-coded in fig 4b and in supp figures 4 d,e.

In figure 5, it would be helpful if the legend stated that Δ MDA was expressing pilE_{SD}. Δ MDA is a population of bacteria that express various PilE proteins. This is now mentioned in the legend of figure 5 (lines 809-810).

In figure 6, panels B and D. Were the PCRs performed on the three independent experiments or 3 times on the same experiment. PCRs performed on the three independent experiments. This has been added in the results section (line 801).

In supplemental figure 2 it would be helpful to have the DNA sequence of the PilE z1-4 strains. We have added these sequences to the supp Figure 2.

In supplemental figure 4, the colors of the first three columns are white. Legend needs to explain this. We have modified the text accordingly (lines 903-905).

In the table of strains, the construction of the Δ MDA strain has the wrong reference. It should be J Exp Med (2005) 201 (12): 1905–1913.

We have modified the text accordingly.

Reviewer #2 (Remarks to the Author):

L233-234 “PilC1 Δ 3-843 allowed transduction and transformation at a level equivalent to 10% of the control (Fig. 3b,c).”

Isn't this an indication that the C domain of PilC may be involved in binding of MDA?

We agree with the reviewer #2 comment that the text was misleading. We have now discussed this point in the results section (lines 204-206): “Interestingly, the truncated protein PilC1 Δ 3-843 allowed transduction and transformation at a level equivalent to 10% of the control while truncated PilC1 Δ 4-977 has lost both transduction and transformation competency (Fig. 3b,c). This suggest that PilC1 Δ 3-843 and PilC1 Δ 4-977 do not complement for piliation and cannot be included in our analysis.” And in the discussion section (lines 376-377): “Although unlikely, we may not exclude that the phage binds with the very last part of the C-terminal domain of PilC”.

L242 Supplementary Fig. 3b: This figure describes bacterial adenylate cyclase two-component assay which is here used to rule out the interactions of Orf6 with minor pilins. Unfortunately, there is no appropriate positive control showing that the Orf6 fusions used in this experiment have folded into a functional form. This is crucial because the type of two-hybrid system used in this experiment means that the proteins have to fold correctly in the cytoplasm. Given that the tested proteins fold in the periplasm prior to assembly into the phage/pili, and have S-S bridges that are required for their correct folding (which do not form in the cytoplasm), it is quite possible that none of the pilins or Orf6 in this experiment are folded into functional domains. In the absence of Orf6 and pilin positive control interactions, negative results shown in this figure cannot discern between failed interactions and misfolding of one or both partners. The only positive control is pIII from M13 phage (M13 pIII - TolA interaction gives the only positive signal in the figure). Why was MDA Orf6 interaction with -N.m. TolA not tested in this setup? Why were no interactions with pilE tested, given the manuscript is focused on showing that PilE is the key receptor for MDA?

We understand the reviewer #2 concern. The original text was misleading. The two-hybrid assay was performed in Oxi-blue *E. coli* from 10.1016/j.jbc.2023.105048, which expresses all the enzymes required for S-S bridge formation in the cytosol. We have now mentioned this point in the text (line 429). Furthermore, although we agree with the reviewer regarding the positive control, we could not have performed this control as TolA is not present in meningococci and the target of ORF6 is not known in *N. meningitidis*.

We agree that this experiment has this drawback and that we cannot assess the folding of ORF6, even though *E. coli* has previously been used to express minor pilin TcpB for structural studies (ref. Gutierrez-Rodarte et al. <https://doi.org/10.1074/jbc.RA119.009980>) and the ORF6 ortholog pIII_{CTX}. We have now clearly mentioned this drawback in the results and softened our conclusion in the text (line 217).

We did not test the interaction between PilE and ORF4 because these two proteins must be assembled into polymers. We have now clearly mentioned that in the results (lines 224-226).

L250-251 + Fig. 4. “we first performed transmission electron microscopy (TEM) on whole bacteria incubated with phages “ The TEM shows MDA longitudinally associated with the pilus, through pVIII-pilE interactions. This may be a secondary step after the initial binding of Orf6 to either pilE or minor pilins, or both, and may strongly contribute to the overall binding. Competition studies using purified MDA Orf6 or MDA phage that has truncated pIII (without the host-binding domains) are required to resolve the pairing i.e. does pIII first interact with -pilC(C-terminal domain), then pVIII interacts with pilE? or Orf6-pilE concurrently with pVIII-pilE?

Reviewer #2 raises an important point. We have now included a MDA binding assay with a phage deleted for *orf6* (Supp figure 5). We see no role for ORF6 in the interaction between the phage and the fiber. The phage deleted for *orf6* exhibits the same behaviour as the wild-type

phage. We have added this result in the result section (lines 263–268, Supp Fig.5), the method section (lines 450 and 651) and we modified the discussion section (lines 389-390).

L251-257/ Fig. 4a; Supplementary Fig. 4: Why is the negative control shown in supplementary figure? Side by side comparison would be more impactful. Was there any statistical analysis of gold bead pilus-association done on multiple cells? This would strengthen the validity of observations.

We have added the control in figure 4a. We have not quantified the quantity of gold beads since beads are virtually absent from our control experiment. Therefore, we have chosen to quantify this interaction through the MDA binding assay (Figure 4b).

Fig. 6-7: Q - why are the values shown in 6c and 6f-g relative (% relative to the value of the control sample), while absolute value is shown in 7b?

Figure 7 was designed as a competition experiment showing that the proportion of pilS3-expressing population increased within the total population. In contrast, Figure 6 indicates the transduction frequency of adhesion on cells compared to the control condition.

L461 “10% NaCl and 20% polyethylene glycol 6000” – Is this the stock concentration or the final concentration? This is way too high as a final concentration for precipitation (normally it is 0.5 M NaCl, 5% PEG for filamentous phages). If only one round of PEG precipitation was done and no further purification steps were performed, the samples would be highly contaminated with host proteins and cellular debris or vesicles such as OMVs.

We apologise for this error in the Methods section. This has now been corrected (line 455).

462-463 “resuspended in PBS 1X (with CaCl₂ and MgCl₂; Gibco) and used directly” In the presence of divalent cations OM fragments or vesicles will not be soluble. Was the insoluble pellet removed from the phage stock by centrifugation, to eliminate OMVs?

We apologise for this oversight and we thank Reviewer #2 for pointing this out. A centrifugation step is performed at the end of the phage preparation. We have added this in the Methods section (line 457-458).

L467-468 “Two hundred microliters of the bacterial suspension were aliquoted into a 24-well plate and mixed with 30 μ l of the MDA phage preparation.” Were the phage stocks titrated? How many phages were used per transduction? What was the m.o.i.? Were there any attempts to quantify phages by densitometry or spectrophotometry? This is also essential to ensure that the same amount of phages is always used between different experiments. See Waldor and Mekalanos 1996 <https://www.science.org/doi/10.1126/science.272.5270.1910> (Table 1) and publications thereafter.

Indeed we did not mention the quantity of phages per μ l, although the transduction frequency was mentioned in lines 482-484 “The efficacy of the phage preparation was verified by transducing 30 μ l of the phage preparation into the Δ MDA strain and was validated if the transduction frequency was greater than 10^{-3} ”. We have now added the number of phages per μ l of phage preparation (assessed by qPCR) (lines 461-470). Considering this value, we can assume that the transduction efficiency is 10^{-3} when 30 μ l of phages are mixed with 200 μ l of bacteria resuspended at an OD₆₀₀ of 1 in transduction liquid medium (M.O.I. = 3).

Same below: L475-479 “The efficacy of the phage preparation was verified by transducing 30 μ l of the phage preparation into the Δ MDA strain and validated if the transduction frequency was greater than 10^{-3} . If no bacteria were obtained at the least diluted point on the agar plates, a CFU of 0.9 was arbitrarily assigned and the point was marked as below the detection limit. Same comment as above: how many phages were used, i.e. what was the m.o.i.?

Please see previous answer.

L607-609 Quantification of pili by western blotting: What was used to plot a standard curve here? Standard should be serial dilutions of a purified pilin subunit quantified by a colorimetric of other absolute

protein quantification method. Serial dilutions of quantified pilin subunits should be run on the same SDS-PAGE gel along with the pili samples, transferred and used to construct a standard curve so that the extracted pilin bands in the westerns can be correctly quantified.

We understand that our method of quantifying pili was misleading for the three reviewers. It is based on a method of pili extraction and purification that was previously published in PNAS in 2021 (<https://doi.org/10.1073/pnas.2109364118>). We agree that the sensitivity of this assay is very low. Consequently, the background signal appeared to be at levels up to 5% of those observed in the wild type. As suggested by reviewer #1, we have proposed a new strategy for pili quantification based on imaging of assembled pili: Imhaus and Duménil (<https://doi.org/10.15252/emj.201488031>). Unlike the method proposed by Imhaus et al., we did not quantify the number of pili, which requires the observation of isolated bacteria, but rather the total surface area of pili detected by immunofluorescence, normalised to the surface area of bacterial DNA detected by DAPI. Using this strategy, we demonstrate that the *pilC1/pilC2* double mutant exhibits piliation similar to that observed in *pilE* or *pilQ* deficient mutants – which was expected. Figure 1c and its legend have been modified accordingly (lines 739-755). The material and method used to construct the strains used for immunofluorescence and the quantification itself have been added (lines 582-585 and 642-646).

L612 “Phage precipitation” method This looks like a protocol for detecting bacteria-associated vs. free phages. The assay commences by mixing phage and bacteria at a huge excess of phages (10E11 phages per 2x10E8; i.e. m.o.i. 500). Since the efficiency of transduction is well below 100% based on data in the manuscript (albeit absolute numbers are mostly missing), this assay where there is huge excess of phages over the host bacteria is puzzling. It is good to see that the phages were quantified before use in this experiment. Method of quantification, however, was not described. Titer vs. quantity of the phages seem to be strongly dependent on the strain, hence it is really important to measure the number of phage particles (quantity of phage) in addition to titration based on host infection. Given the very crude MDA phage purification method is described earlier in the methods section, which results in phage preparations containing large amount of contaminating proteins derived from the host (L451-463; also see a comment above), it would not be advisable to use simple spectroscopy to quantify phages. Instead, quantification of phage DNA would be a much more accurate measure of the number of phages per volume of the sample. Methodology for phage DNA quantification is well-established in the phage literature (densitometry or qPCR).

We agree with Reviewer #2. MDA Φ preparation have been indeed quantified by qPCR. This has now been mentioned in the Materials section (lines 461-470) and we apologise for this oversight. The protocol of the assay of mixing phage and bacteria was actually designed to assess the binding of MDA to pili. To this end, a $\Delta pilT$ mutant was chosen to avoid phage entry.

Minor comments

L35-36 F pilus is not a type IV pilus. Rewrite the sentence: The paradigm of filamentous phage infection has been defined for the phages Ff and CTX, where binding to the tips of bacterial type IV pili is a means of infecting their bacterial host.

We have rewritten the abstract to shorten it and ensure that it meets Nature Communications' editorial requirements. We have, of course, taken the reviewer's pertinent comment into account (19-31).

L42-44 Taken together, this study reveals how the propagation strategy of a filamentous phage, aimed at selecting the best coloniser as a host, can promote the selection of hyperadhesive bacterial variants, linking phage infection to bacterial virulence. Convolutated sentence. Simplify.

We have rewritten the abstract to shorten it and ensure that it meets Nature Communications' editorial requirements. We have, of course, taken the reviewer's pertinent comment into account (19-31).

L62-63 The first filamentous phage of this family was described by Peter H. Hofschneider in *Escherichia coli* in the 1960s and named M13 (Ray et al., 1966). This is not the first report, look for earlier references for Ff Hofschneider PH. 1963. Untersuchungen uber kleine E. coli K 12 bakteriofagen 1 und 2 mitteilung. Z Naturforsch Pt B 18: 203–210. doi:10.1515/znb-1963-0306 Loeb T. 1960. Isolation of a bacteriophage specific for the F+ and Hfr mating types of Escherichia coli K-12. Science 131: 932–933. doi:10.1126/science.131.3404.932 Marvin DA, Hoffmann-Berling H. 1963. A Fibrous DNA phage (Fd) and a spherical RNA phage (Fr) specific for male strains of E. coli ii. Physical characteristics. Z Naturforsch B 18: 884–893. doi:10.1515/znb-1963-1106

L39 We thank the reviewer for this comment and we have added the references in the text.

L63 replace “famous” with “well-known”

L40 We have modified the text accordingly.

L64 replace “CTX” with “CTX ϕ ”

L41 We have modified the text accordingly

L66 “Their main features are the mutually beneficial trait conferred to their host” Ff are not beneficial to the host. They slow the cell cycle and cause envelope stress. Rewrite.

L42 We have modified the text accordingly.

L68-75 Paragraph reads as if only Pf4, CTX and MDA filamentous phages have been reported to have an effect of bacterial host. However, there are a number of filamentous phages that infect *Pseudomonas*, *Vibrio* and plant pathogenic bacteria such as *Xanthomonas* etc. that have been reported to have impact on bacterial virulence.

L51-52 We have modified the text accordingly.

L76-L77 “Almost all filamentous phages described so far infect diderm bacteria and must therefore cross both the outer and inner membranes to infect their target.” This is no longer correct. There are over 10,000 filamentous phage-derived prophages the current sequence databases. Many gram-positive bacteria host filamentous phages, as well as some archaea. See Roux S, et al. 2019. Cryptic inoviruses revealed as pervasive in bacteria and archaea across Earth’s biomes. Nat Microbiol 4: 1895–1906. doi:10.1038/s41564-019-0510-x (and the follow-up articles). Rewrite: Filamentous phages that infect diderm bacteria must cross both the outer and inner membranes to infect their target.

L53 We thank the reviewer for this comment and we have modified the text accordingly.

L80 “such as Ff and related phages that use the sexual/conjugative pilus F of *E. coli* (Hay and Lithgow, 2019).” Cited paper is a general review on Ff phages. There are much more appropriate primary research references for this. Use more appropriate references.

L57 We have added a more appropriate reference.

L86-87 “pilus porin” Type IV pilus assembly system does not have a porin. Its outer membrane component pilQ is a channel of the secretin family that is unrelated to porins. Replace with the correct term.

L63 We have modified the text accordingly.

L118-119, comment: same as above. PilQ is not porin, it is a channel of the secretin family.

L91 We have modified the text accordingly.

L138 – comment: binding of filamentous phages to the tip of the pilus is not a paradigm. For example, filamentous phages infect unpiliated bacteria (e.g. *Yersinia pestis* filamentous phage CUS-1–<https://www.sciencedirect.com/science/article/pii/S0042682210005064>)

L105 We have modified the text accordingly.

L160-161 “Transduction was impaired in mutants with low pili quantity and low transformation efficiency, with the exception of the Δ pilT mutant, confirming...” Unclear sentence. It could be read as if transduction is not impaired in Δ pilT mutants. Rewrite.

L133 We have modified the text accordingly.

L165 “these proteins are poorly involved in phage transduction.” Rewrite: “these proteins have a minor role in phage transduction”

Additional comment: Transduction is used as an assay for MDA infection, so can be referred to as infection.

L137 We thank the reviewer for this comment and have modified the text accordingly.

L203-204 “To confirm that antigenic variation in the Δ MDA pilESA strain is sufficient to explain transduction by MDA Φ , we constructed...”

L17 We have modified the text accordingly.

L220-222 “The near absence of phage infection in the Δ MDA mutS+ Δ recA pilESA strain seems to disqualify the pili tip as an MDA Φ receptor, although it is considered to be the receptor for filamentous phages f1 and CTX (Jacobson, 1972; Gutierrez-Rodarte et al., 2019).” Clumsy sentence (“disqualify”); repetition the statement about the pilus tip is unnecessary. Rewrite.

L192-193 We have modified the text accordingly.

230 Spell out N-terminal domain, or define “Nter”. In the figures there is also a term “N-ter”.

L196 We have modified the text accordingly.

231-232 “Deletion of more than the first 555 amino acids reduced ...” Change to Increasing deletion to 3-843 residues reduced ...

L201 We have modified the text accordingly.

Throughout: Subscripts and superscripts were not included in chemical formulae: MgCl₂, MgSO₄ – check and correct throughout.

We have modified the text accordingly.

Reviewer #3 (Remarks to the Author) and Reviewer #4 (Remarks to the Author):

1. All of the PilE variants (Figure 5) have higher transduction efficiencies than pilE_{SA}. How representative is the reduced charge of pilE_{SA} in primary isolates?

Reviewer #3 raises a very interesting point. pilE_{SA} was initially isolated from a pilS silent locus of the strain NEM8013: pilE_{SA} corresponds to the pilS2 locus of NEM8013 (Nassif et al., 1993, DOI: 10.1111/j.1365-2958.1993.tb01615.x).

We built a tree (see below) from the pilS sequences of 201 strains based on genome sequences obtained from the pubMLST database, with as little redundancy as possible, and placed the pilE_{SA}, pilE_{SB} and pilE_{SD} in the tree. As the reviewer can see, the variant SA, SB and SD correspond to or are close to common pilS. We did not consider it necessary to include this result in the manuscript so as not to make it too lengthy, but if the reviewer and editor feel it is necessary, we will be happy to add it.

Tree of pilS of 201 strains with class I PilE

2. The authors should list or discuss the T4p morphology of all the mutants; PilE quantity alone does not indicate piliation. Doing this will clarify the connection between pilus biogenesis and phage transduction. For instance, some of the pilin deletions (i.e., pilW) appear to have an intermediate transduction phenotype, which could be due to altered piliation.

We understand that our method of quantifying pili was misleading for the three reviewers. It is based on a method of pili extraction and purification that was previously published in PNAS in 2021 (<https://doi.org/10.1073/pnas.2109364118>). We agree that the sensitivity of this assay is very low. Consequently, the background signal appeared to be at levels up to 5% of those observed in the wild type. As suggested by reviewer #1, we have proposed a new strategy for pili quantification based on imaging of assembled pili: Imhaus and Duménil

(<https://doi.org/10.15252/embj.201488031>). Unlike the method proposed by Imhaus et al., we did not quantify the number of pili, which requires the observation of isolated bacteria, but rather the total surface area of pili detected by immunofluorescence, normalised to the surface area of bacterial DNA detected by DAPI. Using this strategy, we demonstrate that the *pilC1/pilC2* double mutant exhibits piliation similar to that observed in *pilE* or *pilQ* deficient mutants - which was expected. Figure 1c and its legend have been modified accordingly (lines 739-755). The material and method used to construct the strains used for immunofluorescence and the quantification itself have been added (lines 582-585 and 642-646).

3. Others have shown that *tfpC* deletion in *N. meningitidis* and *N. gonorrhoeae* affects Tfp biogenesis, pilin quantity, and transformation efficiency (Muir et al. Nature Communications, 2020; Hu et al. mBio, 2020). In this study, a $\Delta tfpC$ mutant displayed no difference in transformation efficiency and pilin quantity. The authors should discuss this discrepancy.

In Hu *et al.*, the protein TfpC is described as being involved in maintaining the extended state of pili, but not as being necessary for pili expression. In Muir *et al.*, however, bacteria deleted for *tfpC* were found to be piliated at 31% the rate of the wild-type strain, and there was a decrease of more than one log in the transformation frequency. We constructed two types of *tfpC* mutant: an insertion mutant with a transposon introduced into the *tfpC* gene, and a deletion mutant with the *tfpC* gene replaced by a resistance cassette. In both cases, the same phenotypes were obtained: no effect of TfpC inactivation on transformation or transduction, and a very slight reduction in piliation. This was verified in two different strains of *N. meningitidis*, namely the Z5463 strain and the NEM8013 2C4.3 strain.

We have included this point in the discussion (lines 366-371), although we have no explanation for this difference in the obtained phenotype in our hands.

4. While they show that PilE is critical for phage transduction, the authors should be careful not to deemphasize the potential role of the other Tfp components in this process. For example, PilQ may also be important for MDA phage transduction, but its loss also results in non-piliated bacteria. Thus, the loss of piliation could be confounding the effect of *pilQ* deletion in phage transduction.

We have included this point in the discussion section (lines 378-379).

Minor Points:

1. Line 58: Caudovirales should be italicized.

L35 We have modified the text accordingly.

2. Line 66: Please clarify the “mutually beneficial trait”.

L40 We have modified the text accordingly.

3. Line 69: Please provide examples of the environmental signals that trigger phage gene induction.

L45 We have modified the text accordingly.

4. Line 140: Please choose a different word from “benefits” or clarify that the phage benefits from the positive charge.

L107 We have modified the text accordingly.

5. Figure 1: List or discuss the piliation states of each mutant.

We add to fig 1a, 1b, and 1c a detection threshold, visualized as a shaded grey area, corresponding to the upper limit of the 95% confidence interval of non-piliated $\Delta pilE$ mutant, so that the mutants can be compared with the unpiliated $\Delta pilE$ mutant.

6. Figure 1: Reconfigure the graphs so that the transduction, transformation, and pilin quantities are all next to each other for each gene deletion. The colors of each bar could be different for each measurement.

We respectfully disagree with reviewer #3. The proposed configuration is too complicated to visualise well, and that the result would be difficult to interpret.

7. Figure 1 A-C needs statistics.

We have added statistical analyses to Figure 1.

8. Lines 205-206: this statement needs a citation.

L177 We have added a citation.

9. Line 215: Re-write this for “better infects certain variant”, as the phage can still infect other pilE variants, just at lower efficiency.

L187 We have rewritten this sentence.

10. Line 228: Why was PilC1 chosen instead of pilC2?

The PilC1 and PilC2 C-terminal domains are very close to each other in *N. meningitidis*. We chose PilC1 based on the molecular tools available in the laboratory.

11. Please speculate why a pilE deletion retains some transduction and transformation efficiency.

It is known that filamentous phages interact with pili, allowing them to be pulled in by the pili secretin (Hay *et al.* EMBO reports 2019, <https://doi.org/10.15252/embr.201847427>). Here, we observed that transduction in the $\Delta pilE$ mutant is 10,000 times lower than in the wild type. The same is true for the $\Delta pilQ$ mutant. Therefore, we can assume that any pili-defective mutant will have a transduction frequency below the detection threshold, and that colonies growing in these mutants will be spectinomycin-resistant colonies due to spontaneous mutation (note that spectinomycin resistance is due to a mutation in the 16S rRNA: <https://doi.org/10.1128/aac.44.5.1365-1366.200>).

12. Line 265: it looks like some phage did bind to PilE_{SA}, just at reduced levels

L242 We agree with the reviewer that MDA Φ may interact with PilE_{SA} at very low level. The sentence has been modified accordingly.

13. Supplemental Figure 4e: What is the MDA binding to in the $\Delta pilE$ strain? It is unclear how they calculated binding. This should be clarified in lines 280-282.

We think that binding to a $\Delta pilE$ strain corresponds to background and non-specific labelling.

We clarified this point in the text (lines 258-259).

14. Supplemental Figure 3: all these experiments test Orf6 binding to the pilus fiber proteins. Why was Orf4 not tested, as this protein was shown to be responsible for PilE binding?

We did not test the interaction between PilE and ORF4 because these two proteins must be assembled into polymers. We have now clearly mentioned this point in the results section (lines 224-226).

15. Figure 5b: clarify that the reference strain (Z Δ MDA) carries the PilESD variant. It is unclear from this graph.

The strain Z Δ MDA corresponds to a mix of *pilE* sequences; no major sequence can be determined. We have added this point in the text to line 127.

16. Line 318-319: Why 70-71 in D region was analyzed is unclear. Are these charges not on the surface? Please justify.

We examined the differences between the amino acid sequences of PilE_{SA} and PilE_{SD}, focusing on position 70-71 and that of the D region. These amino acids are exposed, as shown in supplemental Fig. 6.

17. Figure 6B: The % positive and negative bacteria with the *pilES3* variant should be separate bars. This graph is deceiving because it appears that % *pilES3* positive bacteria are at 50%, when it is only 1%.

We thank reviewer #3 for this comment. We have changed the graph. We now only show the results that correspond to the Y legend.

18. Figure 6: Clarify that for figures A-C, transduction occurs after colonization, while for figures D-F, transduction occurs before colonization

We have added indications on the figure 6a and 6d, with "1-" and "2-" to explain the chronology of the experiments.

19. Figure 6c should have the same Y-axis scale as Figures 6f-g.

We have modified the graph accordingly.

20. Can you comment on how widespread the MDA phage is in *N. meningitidis*?

The presence of an average of two to four prophages per *N. meningitidis* strain indicates that prophages are very common and result from reinfection rather than duplication (Kawai et al., DNA Research 2005, doi: 10.1093/dnares/dsi021, Al Suwayyid et al., GBE 2020, doi:10.1093/gbe/evaa023). We now discuss this point in the text (lines 407-409).

21. Figure 7C: This image suggests that the MDA phage improves transmission, but the data do not necessarily support this.

We have modified the figure to clarify this point.

22. Can MDA phage bind the Class II PilE and infect?

We haven't tested the class II *pilE* sequences, but the available genomes of strains with type II *pilE* have the prophage MDA in their sequence. We have evaluated the charge (pI) of class II *pilE*. It is quite neutral, which should not seem to prevent the phage from attaching to the pili.

23. Is it possible that only a few pili are necessary? Some of these mutants may appear non-piliated but may have a few pili capable of facilitating transduction.

Looking at Figures 1a, 1b and 1c, we can see a strong correlation between the amount of pili and the rates of transformation and transduction. It appears that a small number of pili is sufficient for the bacterium to be infected by the phage, but transduction is more effective when piliation is extensive.

REVIEWER COMMENTS and ANSWERS

Reviewer #1:

In figure 1e, the authors do not define the PilE sequence of ZAMDA (line 179). This should be in supplemental figure 1. Supplemental figure 2 does not have this sequence, or pilEsb, and this also needs to be added here.

We agree with Reviewer #1. However, the pilE gene of ZAMDA cannot be determined, since the strain corresponds to a mix of pilE sequences, with no dominant one having been determined. We have added this information to line 127.

I do not understand this response. The strain has been manipulated in the laboratory to lack phage. This would indicate a selection event. Therefore, even if the parent from the original case was diverse, the mutant should be clonal. This information is needed to understand the relationship between the parent and the mutant pili that they are analyzing. What if the PilD region of the parent is the same as PilESA or PilESB?

Answer: We agree with reviewer #1 that the initial strain was clonal (one colony was picked), but this clone underwent several passages to eliminate the circular cytoplasmic form of MDA. The frequency of antigenic variation of the *pilE* locus with *pilS* loci is known to be 1 to $2 \cdot 10^{-3}$ in *Neisseria meningitidis* (<https://journals.asm.org/doi/10.1128/jb.00465-18>). Under these conditions, the population cannot be clonal for the *pilE* locus. Sanger sequencing of *pilE* of ZAMDA is shown below. It did not allow us to determine the *pilE* sequence. Note that the D-regions are different between all the *pilS* (see supp figure 2). The *pilSA* D-region is not present in any ZAMDA *pilS* sequence.

This is mentioned line 131 in the text and lines 459-463 in the methods section.

 In figure 1, they do not define the limit of detection for a transduction experiment? If *PilE* is truly required for transduction, it would seem that the transductants identified in figure 1a for *PilE* represent the mutation frequency to spectinomycin resistance. Is this true?

We thank the reviewer for this comment. Indeed, it is known that filamentous phages interact with pili, allowing them to be pulled in by the pili secretin (Hay et al., EMBO reports 2019 <https://doi.org/10.15252/embr.201847427>).

This reference does not show that the phage interact with the gonococcal pilin secretin.

Answer: We agree with reviewer#1, our sentence is badly written and we apology for this. We meant to write that 'Filamentous phages interact with pili, allowing them to cross the secretin during pili retraction.' (Hay et al., EMBO Reports, 2019, <https://doi.org/10.15252/embr.201847427>; Jacobson 1972, <https://doi.org/10.1128/jvi.10.4.835-843.1972>; Dixon *et al.* 2016, <https://doi.org/10.1371/journal.ppat.1006109>).

Here, we observed that transduction in the $\Delta pilE$ mutant is 10,000 times lower than in the wild type. The same is true for the $\Delta pilQ$ mutant. Therefore, we can assume that any pili-defective mutant will have a transduction frequency below the detection threshold, and that colonies growing in these mutants will be spectinomycin-resistant colonies due to spontaneous mutation (note that spectinomycin resistance is due to a mutation in the 16S rRNA: <https://doi.org/10.1128/aac.44.5.1365-1366.200>). We have therefore added the background threshold to the figure 1.

The establishment of background is based on a series of assumptions. If they are transductants, they should have the spec cassette. They should use this to determine the detection level.

Answer: We understand the reviewer#1 concern. We have experimentally evaluated the frequency of spontaneous spectinomycin resistance mutant for $Z\Delta MDA$, $Z\Delta MDA \Delta pilE$, and $Z\Delta MDA \Delta pilQ$ without addition of phage. The results of spontaneous resistance to spectinomycin in the experimental conditions described for figure 1 without phage is $3.2 \times 10^{-8} \pm 4 \times 10^{-8}$ (SD) for $Z\Delta MDA$, $1.3 \times 10^{-8} \pm 2 \times 10^{-8}$ (SD) for $Z\Delta MDA \Delta pilE$ and $3.9 \times 10^{-8} \pm 7 \times 10^{-8}$ (SD) for $Z\Delta MDA \Delta pilQ$ (n=3 experiments in duplicates). Colonies grown on spectinomycin plates for $Z\Delta MDA$ and the $Z\Delta MDA \Delta pilE$ and $Z\Delta MDA \Delta pilQ$ mutants do not

contain spectinomycin-resistance cassette. We have added this information to the legend of figure 1 and the methods.

In figure 2, panel b, when the authors sequenced 10 random colonies, 50% underwent phase variation.

This is much higher than that that has been reported for the meningococcus (doi: 10.1128/JB.00465-18). How do the authors explain this extraordinarily high rate?

We agree with Reviewer #1 and were also surprised. However, this rate must be considered in relation to the number of generations that have passed since the PileSA variant was introduced to the original strain (at least two overnight cultures on plates). We observed the same variant rate with a PileSB variant.

This suggests that the starting cultures had mixed genotypes.

Answer: We fully agree with reviewer#1, please see above our answer to the first question.

Furthermore, the strain Z5463 has a mutS mutation (Omer et al., PLoSOne 2011, doi.org/10.1371/journal.pone.0017145), which may explain the high rate of punctual mutations observed in three strains of the ten strains.

If the authors explanation is correct, why did they see the same pilin mutations and not a variety?

Answer: We agree with reviewer#1 concern. However, among the ten strains that were sequenced before transduction (Figure 2b and below), the five sequences highlighted by the reviewer#1 are all different.

That being said, we do not have a definitive explanation for this concern. We observe that 50% of the bacteria selected to express PileSA variants have a different sequence in their *pilE* gene, which can be explained by the high rate of antigenic variation, a high rate of point mutations, or a yet-to-be-discovered event of variation specific to this locus. Eventually, this variants were not recovered after selection of the transduced population.

In the discussion, the authors try to link phage expression with virulence. It is hard for this reviewer to understand the logic. This relationship may exist the first time a bacterium becomes infected with MDA, but subsequent replication in a host would allow for antigenic variation where the strain no longer expresses the Pile needed for phage infection. Unless the strain is constantly being reinfected with MDA, how would phage infection be linked to virulence. Do the authors have any indication of this?

The reviewer #1 raises an important question. The presence of an average of two to four prophages per *N. meningitidis* strain indicates that prophages result from reinfection rather than duplication (Kawai et al., DNA Research 2005, doi: 10.1093/dnares/dsi021, Al Suwayyid et al., GBE 2020, doi:10.1093/gbe/evaa023) and thus bacteria should be constantly reinfected. We now discuss this point in the text (lines 409-411).

These references do not really address the question. One of the references suggests that phage acquisition is by transformation.

Answer:

Reviewer#1 argued that phage infection would be linked to virulence if the strain is constantly being reinfected with MDA, but asked if the authors have any indication of this point.

We do believe that reinfection occurs. First we now show that the transduction rate is around 10^{-3} to 10^{-4} in our setup and we showed earlier that phages are produced when bacteria grow as biofilm on the top of epithelial cells. Then, multiple prophages in genomes seems to come from reinfection (Kawai et al., DNA Research 2005, doi: 10.1093/dnares/dsi021), which support that reinfection occurs *in vivo*.

The hypothesis of transformation was put forward in JEM (Bille et al. 2005) before the phenomenon of transduction was demonstrated in Microbiology (Meyer et al. 2015). Transformation is highly unlikely due to the number of DUS in the MDA sequence (one DUS per 8 kb), and because the insertion sites of MDA in genomes are all different and there is no possible recombination with the upstream and downstream sequences. About the Al-Suwaydi reference, the authors indeed propose the transformation to explain the presence of the Nf1 phage in certain gonococcal strains. However, this is not relevant to our discussion, which concerns only meningococci and we do believe that this reference address the question.

However, we understand that reviewer #1 is not convinced by the term “virulence.” Since we agree that this term is supported only by hypothesis, we have decided to attenuate our conclusion. We have removed the word “virulence” from the abstract (line 30), the introduction (line 106), and the discussion (lines 380 and 382).

Bille *et al.*, JEM 2005

Figure S2. Insertion sites of Nf1 prophage in the *N. meningitidis* strains Z2491, MC58, FAM18. Al Suwayyid *et al.* GBE2020

Figure S1. Distribution of *Nf1* prophage and *dRS3* repeats in reference *Neisseria meningitidis* genomes Al Suwayyid *et al.* GBE2020

Al Suwayyid *et al.* GBE2020

Reviewer #2 (Remarks to the Author):

Authors have taken some steps to improve the quality of writing and data,

We thank reviewer#2 for this comment.

however manuscript requires more work on both fronts. The most critical issue is that number of MDaphi used in binding and transduction experiments is not given. Rather it was generally stated that purified phage numbers were "10E7 per microliter at average". This gives quite a low phage:bacteria ratio. Given that most binding assay results are given in relative values, the veracity of findings is hard to gauge.

Answer: We understand that the main concern of reviewer #2 is that data expressed as a percentage are less reliable for the reader and that the number of phages used for transduction is not specified.

We have now clarify these points. In the methods, we now indicate that the bacteria:phage ratio is approximately 1:3, line 517. We have modified all the figures to show the transduction rate that is now calculated by dividing the number of CFU obtained on Sp-containing agar plates - which were transduced by the phage - by the total number of bacteria on antibiotic-free agar plates. We have modified figures 1a, 1e, 2e, 3b, 5b, 6c, supp figures 1d, 4b, their legends as well as the methods, and performed the statistical analysis again accordingly. The transduction rate of our control is around 10^{-3} to 10^{-4} , which is consistent with that observed for CTX, for exemple (2.5×10^{-4} for the control, doi: 10.1074/jbc.M117.786061). Please note that the variability is now higher because the data were no longer normalised for transduction in wild-type bacteria in each experiment, although the results and conclusions are the same as in the first version of the manuscript.

With regard to the binding assay shown in Figure 4, binding was calculated based on the staining area of the pili in different images. It is therefore not possible to calculate the number of MDA bindings. We agree that the term “MDA binding” was misleading and we have therefore replaced it with “MDA staining”.

Reviewer #3 (Remarks to the Author): The authors have responded well to the previous three reviews.